# A-Type Natriuretic Peptide Alters the Impact of Azithromycin on Planktonic Culture and on (Monospecies and Binary) Biofilms of Skin Bacteria *Kytococcus schroeteri* and *Staphylococcus aureus*

**DOI:** 10.3390/microorganisms11122965

**Published:** 2023-12-12

**Authors:** Ekaterina V. Diuvenji, Ekaterina D. Nevolina, Ilya D. Solovyev, Marina V. Sukhacheva, Sergey V. Mart’yanov, Aleksandra S. Novikova, Marina V. Zhurina, Vladimir K. Plakunov, Andrei V. Gannesen

**Affiliations:** 1Federal Research Center “Fundamentals of Biotechnology”, Russian Academy of Sciences, 119071 Moscow, Russia; ekaterina.dyvenji@mail.ru (E.V.D.); katia.nevolina@yandex.ru (E.D.N.); sukhacheva@biengi.ac.ru (M.V.S.); semartyan@inbox.ru (S.V.M.); mzhurik@gmail.com (M.V.Z.); yavladimir14@gmail.com (V.K.P.); 2LLC “BAVAR+”, 129626 Moscow, Russia; a.novikova@deepfood.tech

**Keywords:** *Staphylococcus aureus*, *Kytococcus schroeteri*, atrial natriuretic peptides, biofilms, multispecies biofilms, human skin microbiota, hormones, antibiotics, azithromycin

## Abstract

It has been established that the human atrial natriuretic peptide is able to alter the effect of azithromycin on *Kytococcus schroeteri* H01 and *Staphylococcus aureus* 209P monospecies and binary biofilms. The effect of the hormone depends on the surface type and cultivation system, and it may have both enhancing and counteracting effects. The antagonistic effect of the hormone was observed mostly on hydrophobic surfaces, whereas the additive effect was observed on hydrophilic surfaces like glass. Also, the effect of the hormone depends on the antibiotic concentration and bacterial species. The combination of azithromycin and ANP led to an amplification of cell aggregation in biofilms, to the potential increase in matrix synthesis, and to a decrease in *S. aureus* in the binary community. Also, ANP, azithromycin, and their combinations caused the differential expression of genes of resistance to different antibiotics, like macrolides (mostly increasing expression in kytococci), fluoroquinolones, aminoglycosides, and others, in both bacteria.

## 1. Introduction

The human microbiota is a complex “consortium of consortia”, which is permeated with complex relationships between members of the community. Direct and indirect interactions between microorganisms and the human organism sustain the homeostasis of the whole system. The commensal microbiota protects the human organism against pathogens and provides numerous positive effects [1], while the human organism provides habitats, nutrients and protection for its microbiota. The microbiota and human organism are interrelated also by signal molecules, and human hormones, as products of human endocrine system, are a part of this interkingdom signaling. Such an interaction is a field of research in a recently formed discipline—microbial endocrinology [2]. However, different internal and external impacts may shift the healthy balance, and lead to dramatic changes in human health. For instance, incorrect chemotherapy results in the emergence of antibiotic-resistant strains of pathogens, which are very difficult to eradicate and to date are one of the most dangerous threats to global health. On the other hand, it seems to be possible to correct the traditional antibiotic-based chemotherapy of infections using an additional molecules, which are able to increase the antibiotic efficacy. The last is important also because the human microbiota mostly exists in the form of multispecies biofilms—highly structured aggregated communities encapsulated in the extracellular matrix [3]. Biofilms themselves are resistant to antibacterial compounds due to the biofilm phenotype and matrix [3], and the eradication of pathogens inside biofilms is a challenge for medics.

Concerning certain members of human microbiota and namely skin microbiota, *Kytococcus schroeteri* and *Staphylococcus aureus* are microorganisms of interest to the scientists focused on them. The first is a poorly investigated coccus, which did not have a complete genome reference in the NCBI database when the current study was finished. The only draft genome assembly was performed in 2019 for our strain H01 [4]. Scattered data on examples of bacteremia caused by *K. schroeteri* [5] reveal only the fact that this microorganism is able to cause rare infections, mostly in patients with attenuated immunity. However, authors have noticed the resistance of *K. schroeteri* to different classes of antibiotics: beta-lactams, macrolides, and clindamycin. Hence, due to its skin-associated habitat in human organisms [6,7], it must make contact with *S. aureus*—the second skin bacterium studied here and one of the most popular and well-known Gram-positive opportunistic pathogens. This is why *S. aureus* has been chosen as one of several model microorganisms for biofilm formation studies, and a lot of data have been accumulated regarding its molecular machinery, biofilm formation and quorum sensing, antibiotic resistance, etc. *S. aureus* is also becoming a model for artificially reconstructed microbial community studies with Gram-negative bacteria (for instance, [8,9]), Gram-positive bacteria (for example, [10,11]), and yeasts (for example, [12]). Currently, it is also established that *S. aureus* is a part of commensal skin microbiota, especially in the nasal area [13]. Also, *K. schroeteri* is a bacterium potentially possessing the ability to influence the microbial community via its resistance to several classes of antibiotics, and *S. aureus* is a well-known nosocomial infection agent, especially in its methicillin-resistant (MRSA) variants [14]. Because of these facts, we consider the community of *S. aureus* and *K. schroeteri* interesting to investigate. We believe that it is worth being studied due to the almost-absent data on how this community is impacted by human humoral factors, i.e., hormones.

Microbial endocrinology [15] is a relatively new cross-disciplinary area of research dedicated to interconnections between human microbiota and human humoral systems. The recent review by Arif Luqman contains the most up-to-date summary of existing data [16] on how hormones affect microorganisms. However, despite the apparent breadth of research in the field, the data in microbial endocrinology seem to be a set of very disparate results that are difficult to put together, making it challenging to develop any fundamental theories. And because most studies are being conducted on monospecies cultures and biofilms, nothing is known about the regulatory effect of hormones on microbial multispecies biofilms, or even on the simplest communities, binary biofilms. The data obtained recently allow us to suggest that the presence of the second microorganism in the community dramatically shifts the behavior of the first counterpart and the effect of an active compound [17,18,19,20]. Thus, this is the first aspect of importance of these investigations. The second is that the issue of antibiotic tolerance in bacteria is one “of the biggest threats to global health, food security, and development today” [21]. Resistant strains can cause severe infections and increase the death rate around the world; in addition, interactions between resistant and sensitive species may lead to diminished antibiotic efficacy against sensitive strains [22]. However, human hormones can help in solving this problem. Natriuretic peptides (NP) here can be of a special interest. First, they are intensively studied as an important participants of numerous physiological processes in humans, and some time ago the first evidence of NP impacts on bacteria were obtained [23]. After this, C-type natriuretic peptide (CNP) was reported as a regulator for *Pseudomonas aeruginosa* biofilms, and AmiC protein as a potential receptor of CNP in bacterial cells was predicted [24]. Next, it was shown that CNP may have another, AmiC-independent mode of inhibitory action on *P. aeruginosa* biofilms in a higher (1 µM) concentration [25]. The recent study demonstrated that another NP—atrial natriuretic peptide (ANP), is reported to be (i) the dispersion trigger and (ii) an adjuvant for tobramycin, polymyxin B, and imipenem [26] activity against *P. aeruginosa* biofilms. Finally, a very recent study investigating ANP action on *P. aeruginosa* revealed the AmiC agonist osteocrin disperses the *P. aeruginosa* biofilms in the same manner as ANP [27], which proves that AmiC is a sensor of NP in pseudomonas cells.

Concerning the study of the ANP effects on bacterial multispecies biofilms, it was shown that ANP (both with CNP) is able to regulate the communities of skin commensal bacteria *Cutibacterium acnes* and *S. aureus* in a temperature- and concentration-dependent manner [28]. In the same way, mixed-species biofilms of *C. acnes* and *Staphylococcus epidermidis* are regulated by both ANP and CNP [20,29]. Recently, we began an investigation of the community of *S. aureus* and *K. schroeteri* in the presence of ANP, and we found that this hormone actually acts as a microorganism-activated regulatory factor for the community [19]. Hence, taking into account all these facts, we decided to analyze how ANP may possibly affect the activity of azithromycin on *K. schroetery*, *S. aureus* individually and in the binary community. In this study, we focused on the effect of azithromycin on the community and how ANP may shift this effect. Azithromycin is a widely used semi-synthetic macrolide broad-spectrum antibiotic [30], and if *K. schroeteri* were to be reported as a macrolide-resistant bacterium [5] while *S. aureus* is one of the most dangerous nosocomial pathogens (Jaradat et al., 2020) [31], the study of how ANP potentially can shift the azithromycin activity would reveal wide perspectives for future research and practical use.

## 2. Materials and Methods

### 2.1. Strains and Cultivation

*K. schroeteri* H01 was isolated from the skin of a healthy volunteer as described previously, characterized, and stored in the UNIQUEM Collection under the number UQM41482 [4,32]. The bacterium was stored and prepared for the experiments as described elsewhere [19]. Briefly, a colony biomass from the lysogeny broth (LB) agar (Diam, Moscow, Russia) was inoculated into 20 mL of LB, then cultivated for 24 h at 33 °C and 150 rpm aerobically. *S. aureus* 209P was stored and cultivated in the same way. To obtain monospecies cultures and biofilms, the suspension’s optical density (OD) at λ = 540 nm was adjusted to 0.5 using physiological saline (PS). The PS contained 0.9% NaCl (Diam, Moscow, Russia) in the distilled water. To obtain binary cultures and biofilms, the OD_540_ of both bacterial suspensions was adjusted to 1.0; then, the suspensions were mixed in a proportion of 1:1, and the final mixture was inoculated into an appropriate experimental system. During the experiments, bacteria were cultivated in the reinforced clostridial medium, as was described elsewhere [19].

### 2.2. Active Compounds

Azithromycin (Teva Pharmaceutical Industries Ltd., Petah Tikva, Israel) was dissolved in 10 mL of ethanol to a final stock concentration of 50,000 µg/mL, then stored at −18 °C. Then, a new stock solution was prepared. To test appropriate concentrations, a series of dilutions was performed in the RCM medium before administration to the experimental systems. Atrial natriuretic peptide (ANP; Alfa-Aesar, Ward Hill, MA, USA) was diluted in sterile MilliQ water (MQW) to a final stock solution of 1 mg/mL and stored at −18 °C. Before the experiment, a series of dilutions in the sterile MQW were performed and administered in the RCM at the beginning of cultivation.

### 2.3. Growth of Bacteria on the Polytetrafluoroethylene Cubes

The monospecies planktonic cultures and biofilms were cultivated aerobically in liquid RCM medium, where the polytetrafluoroethylene (PTFE) cubes served as a carrier for biofilms. This method has been described elsewhere [33]. Briefly, 21 cubes were covered with 3 mL of the RCM in 22 mL glass screw-cap tubes and sterilized at 105–110 °C for 30 min. An appropriate volume of an azithromycin solution was added into the sterile medium, and afterward, 50 µL of a bacterial monospecies culture was inoculated. We tested the effect of the antibiotic on a wide range of concentrations: 0.001; 0.01; 0.05; 0.1; 0.5; 1; 2; 4; 6; 8; 10; 20; and 40 µg/mL. Planktonic cultures and biofilms were cultivated for 24 h, 48 h, and 72 h aerobically at 33 °C and 150 rpm. After the incubation, the OD_540_ of the planktonic cultures was measured. Biofilms were stained and washed from the planktonic suspensions, fixed with ethanol, and stained with 0.1% crystal violet (CV, Sigma, Burlington, MA, USA); then, and the OD_590_ of the CV extracts was measured. To test the action of ANP in combination with azithromycin, the hormone was added into the medium for a final concentration of 6.5 × 10^−10^ M. This concentration had previously been established as active [19].

### 2.4. Growth of Biofilms on the Glass Microfiber Filters

The monospecies and binary biofilms were cultivated on glass microfiber filters (GMFF, Sigma, Burlington, MA, USA) in liquid RCM and on the surface of the solid RCM medium, as has been described elsewhere [18]. Cultivation on the solid medium provides a reduction in the initial adhesion time, while in the liquid RCM, the initial adhesion stage was fully presented. Hence, these two models can allow us to distinguish whether an active compound affects the adhesion. Briefly, in the solid-medium system, 20 mL of agar RCM with or without the addition of active compounds was placed into Petri dishes, when the sterile glass filters 20 × 20 mm were placed onto the surface (two filters per culture), and 20 µL of monospecies or binary suspensions was inoculated in the center of the filter. In the second liquid-medium system, 3 mL of the RCM was added into the Balch tubes and sterilized at 105–110 °C for 30 min. Then, two filters were submerged in the medium, active compounds were administered, and 50 µL of an appropriate culture was inoculated. Plates and tubes were incubated aerobically at 33 °C for 24, 48, or 72 h. For each bacterium or binary community, one filter was physically homogenized in the sterile PS, and the resulting suspension was diluted and plated onto LB-agar Petri dishes for CFU counting. The second filter was stained with 0.1% solution of 3-(4,5-Dimethyl-2-thiazolyl)-2,5-diphenyl-2H-tetrazolium bromide or MTT (Diam, Moscow, Russia) for a metabolic activity test. MTT was diluted in the sterile LB. Filters were transferred (one per a well) to six-well immunological plates (Wuxi NEST Biotechnology, Wuxi, China), and 3 mL of the MTT solution was added to each well and incubated for 30 min at room temperature (RT). Then, filters were soaked with paper towels and washed with the distilled water, and formazan was extracted in 3 mL of dimethyl sulfoxide (Ekos-1, Moscow, Russia). In parallel, for each filter suspension after the homogenization, the samples were fixed on the microscope’s glass slides, fixed with flame, and stained with 0.1% CV to analyze the aggregates of the cells in the suspension. Aggregation was evaluated using light microscopy (Zeiss Jena, Jena, Germany) on the magnification x900 with oil immersion. The average amount of cells per aggregate was calculated, and the ratio of single cells to aggregates was evaluated.

### 2.5. Growth Kinetics Study

To investigate how the active compounds would affect the growth parameters in monospecies and binary cultures of *S. aureus* and *K. schroeteri*, the OD_540_ was measured every 15 min in 96-well microtiter plates (TPP, Trasadingen, Switzerland). The cultivation was performed as described previously [18]. Briefly, as was described for systems with GMFFs, two types of inoculation systems were used. In the first system, cultures were prepared in the LB medium as described previously, then washed twice with sterile PS. The OD_540_ was adjusted to appropriate levels with the PS, and monospecies and binary suspensions were prepared. Afterwards, the cultures were inoculated into the wells (200 µL of a suspension per well) and incubated for two hours at RT to allow the cells to adhere to the bottom of the well. Subsequently, the suspensions were gently removed, and the wells were rinsed with sterile PS to remove the residual unadhered cells. Then, 200 µL of liquid RCM, either containing the active compounds or without them, was added to the wells. In this system, cultures grew from an adhered state, and the biofilms prevailed in their contribution to the OD_540_ in comparison with planktonic cells. In the second system, the liquid RCM was added to the wells (200 µL per well); then, 3.3 µL of an appropriate suspension was inoculated into each well. Hence, the growth began from the suspension, and planktonic cultures were more represented in comparison with the biofilms. Cultures were incubated for 72 h at 33 °C and 150 rpm using an XMark spectrophotometer (Bio-Rad, Hercules, CA, USA). Growth curves were obtained using Microsoft Excel 2010 software (14.0.7268.5000 64X). In each repeat of the experiment, the specific growth rate was calculated as µ = (ln(OD_2_/OD_1_))/(t_2_ − t_1_), where OD_1_ and OD_2_ (and the corresponding *t* meanings) represented the beginning and end of the linear portion of the semi-logarithmic plot of the growth curve, respectively. Semi-logarithmic plots were constructed, and the linear portions were calculated using the SLOPE function in Excel. SLOPE allowed us to find the coefficient “a” of linear regression y = ax + b; thus, we identified the portion as “linear” when (i) there was a minimal standard deviation of coefficient a, and (ii) a was maximal; that is, the curve was closer to vertical. Furthermore, the generation time *t* was calculated using the formula *t* = ln(2)/µ.

### 2.6. Confocal Microscopy and Fluorescence Hybridization In Situ

Samples for confocal laser scanning microscopy (CLSM) were prepared as described previously [19]. The fluorescence hybridization in situ (FISH) was used to detect cells of *S. aureus* in binary communities. The probe used for *S. aureus* was 5′-GAA-GCA-AGC-TTC-TCG-TCC-G-3′ (Syntol, Moscow, Russia), labeled with rhodamine R6G [34]. Monospecies and binary biofilms were grown in glass-bottom 24-well plates (Thermo Fisher, Waltham, MA, USA). The RCM medium, with or without the active compounds or their combinations, was dispensed into the wells (one mL per well); 17 µL of the prepared cultures of *K. schroeteri*, *S. aureus,* or the binary communities were then inoculated, and the plates were incubated at 33 °C and 150 rpm for 24 h and 48 h at 33 °C and 150 rpm aerobically. After the incubation, the biofilms were fixed with 1 mL per well of 96% ethanol for 20 min at the RT. Then, ethanol was removed, and the *S. aureus*-containing biofilms were hybridized as described previously [19], with modifications in the composition of the hybridization buffer required for the probe which was used [24]. After the final washing, all samples were additionally stained with SYTO9 Green (Thermo, Waltham, MA, USA) for 15 min as described previously [18]. The washed samples were dried and covered with Prolong Gold antifade mountant (Thermo Fisher, Waltham, MA, USA) to avoid rapid quenching of the fluorescence. The plates were wrapped with aluminum foil and stored in the dark at 4 °C.

A blind analysis of the samples was carried out and 3D images were obtained using a Nikon TE2000-U microscope (Nikon, Tokyo, Japan) with oil Nikon 100x 1.3NA S Fluo objective and a Becker&Hickl DCS-120 scanning confocal system for image acquisition. Fianium WhiteLase SC-480-6, with an AOTF filter, was used for fluorescence excitation, in addition to Becker&Hickl PMC-100-1 detectors for signal acquisition (Becker&Hickl GmbH, Berlin, Germany). Green channel—470 nm excitation light and HQ495LP Chroma (Becker&Hickl, Berlin, Germany) + 509/22 BrightLine Semrock (Fairport, New York, NY, USA) emission filters—and red channel—530 nm excitation, and HQ550LP Chroma (Becker&Hickl, Berlin, Germany)+ 480bp40 Omega (Omega Optical, Brattleboro, VT, USA)— emission filters were employed. For z-scanning, we upgraded the manual z-knob using a self-made electronic focusing system. Standard Becker&Hickl SPCM software (version 9.89) [35] was utilized for image acquisition. A typical image had 1024 × 1024 (176.6 × 176.6 microns) resolution, z-step—0.5 microns. Raw data were converted into ome.tiff image stacks by self-written Python scripts using the NumPy (version 1.26.0) [36], sdtfile, and tifffile (v2022.5.4) [37] libraries.

The resulting OME-TIFF files were analyzed using the Comstat2 (TDU, Copenhagen, Denmark) [38,39] plug-in of the ImageJ package software (1.48v, Java 1.6.0_20 (64-bit), NIH, Bethesda, MD, USA) based on predesigned computational algorithms). For each well, at least 5 3D raw files were taken for quantitative analysis. Two parameters were determined: the average biomass density per area unit (μm^3^/μm^2^) and total pixel amount in each sample. Three-dimensional images were made from OME-TIFF z-stacks using the Volume Viewer 2.0 plugin in ImageJ.

### 2.7. Quantitative PCR

To investigate potential changes in the expression of genes encoding resistance to antibiotics, qPCR was conducted. We analyzed not only the shifts in expression of macrolide resistance genes, but also potential alterations in the expression of genes resistant to other antibiotics. The online service “The Comprehensive Antibiotic Resistance Database” [40] was used to find the genes of resistance in each microorganism. For *K. schroeteri*, we used the closer species *Kytococcus sedentarius* DSM 20547 (Accession # GCA_000023925.1) due to the lack of a complete genome sequence of *K. schroeteri*. Newly extracted total RNA samples were obtained from three independent experiments, as described above. First-strand complementary DNA (cDNA) synthesis for real-time qPCR was performed using Moloney Mouse Leukemia Virus reverse transcriptase according to the manufacturer’s protocol (Evrogen, Moscow, Russia). Specific primers (Table 1) were applied for the synthesis of unique cDNA fragments. At least three pairs of primers for each gene with differential expression were selected using Unipro UGENE v.38.1 [41] using the built-in Primer3 module and the primer selection function for qPCR. The primers were checked in silico using the web resource insilico [42]. To find the optimal primer pairs, hybridization with the total DNA of *K. schroeteri* and *S. aureus* was performed once before the qPCR experiments. Total DNA was extracted from 24 h suspension cultures using a Wizard^®^ Genomic DNA Purification Kit (Promega, Madison, WI, USA). For improved cell wall disruption, the pellet was frozen with liquid nitrogen and milled with glass, as described for the total RNA extraction protocol below.

RNA isolation was performed as described previously [29]. Briefly, Petri dishes containing RCM-agar with and without the addition of active compounds were prepared. One sterile Ø 21 mm GMFF per dish was placed onto the agar surface. The suspensions of *K. schroeteri* and *S. aureus* were prepared as described above, and 25 μL of this suspension was inoculated in the center of each filter. Biofilms were grown for 24 h aerobically at 33 °C. A Magen HiPure Total RNA Kit (Magen, Guangzhou, China) was used for total RNA extraction. The manufacturer’s protocol was utilized, with changes. After incubation, the filter with the biomass was placed into the porcelain mortar. Twenty-five microliters of RTL buffer (with the addition of mercaptoethanol (Sigma, St. Louis, MO, USA) according to the manufacturer’s protocol) was applied onto the biofilm, and 0.5 cm^3^ of crushed glass was placed onto the filter. The mortar was filled top to bottom with liquid N_2_ [43], and the pestle was also cooled in liquid N_2_. When 3/4 of the N_2_ volume was evaporated, the mass in the mortar was vigorously smashed using the pestle until the N_2_ was completely gone and before the moment of ice melting. The cycle involving the N_2_ addition and smashing was then repeated four times. Ultimately, one mL of RTL buffer was added to the ice-cold mortar with the resulting frozen homogenous powder of glass with disrupted cells, and the mass was vigorously mixed until the ice had melted. Then, 1 mL of the suspension was transferred into a sterile 2 mL microcentrifuge tube (Eppendorf, Hamburg, Germany), and the glass powder was pelleted in an Eppendorf Minispin centrifuge (Eppendorf, Hamburg, Germany) at 11,700× *g*, at room temperature, for 15 s. The supernatant was transferred into RNeasy columns, and all subsequent manipulations were conducted according to the manufacturer’s protocol. Agarose gel electrophoresis was performed to check the quality of the total RNA extracted from biofilms. A 1% agarose (Sigma, St. Louis, MO, USA) gel was used, and 0.01% *v*/*v* ethidium bromide (Sigma, St. Louis, MO, USA) was inoculated into the gel. For each experiment, fresh 1X TAE buffer was prepared to fill the electrophoresis cell. The RNA samples were separated at 65 V for 70 min. Ribosomal RNA bands were the main marker of successful extraction, and other RNAs were visualized using a Bio–Rad Gel Doc XR System w/Universal Hood II (Bio–Rad, Hercules, CA, USA) and Gel Doc XR software Quantity One 4.6.3. The samples of RNA were stored at −80 °C.

q-PCR was performed in PB PCR buffer (Syntol, Moscow, Russia) in the presence of SYBR Green I and the passive reference dye ROX (Syntol, Moscow, Russia) for fluorescent signal normalization. For each sample, detection was conducted twice. ddH2O (Syntol, Moscow, Russia) was used as a negative control. Amplification was carried out using the CFX96 Touch™ RT–PCR detection system (Bio-Rad, Hercules, CA, USA) for the following reaction regime: polymerase activation for 5 min at 95 °C, followed by 40 cycles of 15 s at 95 °C–20 s at 55 °C–40 s at 62 °C. The differential expression of the selected genes was measured in comparison to a non-processed control sample. The average means of the target genes were normalized in comparison with the reference genes accession # NR_118997.2 NR_028935 (16S rRNA of *S. aureus* and *K. schroeteri*, respectively). The amount of a target normalized to an endogenic control and a calibrator was determined using the Ct (ΔΔCt) comparison method with the formula 2^−ΔΔCt^. Data analysis was performed using CFX ManangerTM software v. 1.6.

### 2.8. Data Processing and Statistics

All of the experiments were conducted at least in triplicate. The resulting data were processed in GraphPad Prism 8.3.0. (GraphPad Software Inc., Boston, MA, USA). The data were presented as histograms with medians, and means where it was appropriate. The median-based histograms were supported with all data points plotted and error bars from minimum to maximum sample values. Mean-based histograms (cell aggregate size), CLSM files processing were supported by standard deviation error bars. Growth curves were plotted, and the growth parameters were calculated using Microsoft EXCEL 2010 (14.0.7268.5000 64-bit, Microsoft, Redmond, WA, USA). The non-parametric Mann–Whitney U-test was used to evaluate the statistical significance of differences between samples.

## 3. Results

### 3.1. Test of Azithromycin Concentration Range and Work Concentration Selections

First, we selected work concentrations of azithromycin for further experiments. Because of the necessity to investigate the simultaneous effects of azithromycin and ANP and interactions between *S. aureus* and *K. schroeteri*, we decided to take concentrations with statistically significant but quantitatively lower effects. The values should provide an ability to find any changes caused by an addition of a hormone or presence of another microorganism in the binary biofilm. Hence, the search should be conducted mostly in range of subinhibitory concentrations. Also, in lower concentrations, antibiotics may play a signal role and regulate the bacterial behavior [44,45]. Next, subinhibitory concentrations are of special interest because of their frequent appearance during incorrect chemotherapy of infections. Hence, in lower concentrations, antibiotics should be studied as well as in traditional higher concentrations.

After incubation for 24 h, 48 h, and 72 h, the effect of azithromycin on planktonic cultures and biofilms was different depending on time (Figure 1 and Figure 2). *S. aureus* was generally more susceptible to the antibiotic than *K. schroteri*, especially after 24 h of incubation (Figure 1A), when 4 µg/mL of azithromycin inhibited the growth by 67%. At the same time, 2 µg/mL azithromycin inhibited the planktonic growth by only 15%. In concentrations higher than 4 µg/mL, the planktonic growth of *S. aureus* was very low, and was actually near the lower detection threshold. Staphylococcal biofilms, after 24 h of incubation, were slightly more resistant to the antibiotic—in the presence of 4 µg/mL, the inhibition was 33%; however, 6 µg/mL azithromycin was effective in planktonic cultures (inhibition by 70%).

An increase in incubation time to 48 h led to a shift in the response of *S. aureus* to azithromycin. First, despite the concentration of 4 µg/mL remaining as the threshold of strong inhibition, the planktonic cultures at this concentration were inhibited by 89%, which is actually close to the lower detection limit (Figure 1C). Biofilms were inhibited in the same manner (Figure 1D); however, at lower concentrations (namely, 0.5, 1, and 2 µg/mL), azithromycin stimulated the growth of *S. aureus* biofilms (by 30, 24, and 44%, respectively). After 72 h of incubation, the inhibitory effect of azithromycin was decreased (Figure 1E,F). The strong inhibition threshold was at a concentration of 8 µg/mL in the case of 72 h biofilms of *S. aureus*. At 4 µg/mL, the inhibition was 45% in the case of planktonic cultures. The biofilms were not sensitive to the antibiotics at this concentration.

Therefore, we decided to use further the concentration to 4 µg/mL of azithromycin because of partial inhibiting of *S. aureus* growth in most cases (in particular in case of 24 h and 28 h biofilms).

In the case of kytococci, the sensibility of planktonic cultures after 24 h of incubation (Figure 2A) was generally similar to that of *S. aureus*; however, the inhibition effect increased more smoothly than in the case of staphylococci. A concentration of 4 µg/mL of azithromycin inhibited the planktonic growth of *K. schroeteri* by 66%; however, residual growth was detected at antibiotic concentrations up to 40 µg/mL. The biofilms of kytococci after 24 h of incubation seemed to be in the initial growth stage; hence, they were potentially too weak and unstable, resulting in high fluctuations (Figure 2B). However, the median values reflect the relative resistance to the antibiotic at all concentrations. A decrease in biofilm growth was observed at 6 µg/mL of azithromycin (by 49%), and the inhibitory effect fluctuated around this value up to a concentration of 40 μg/mL.

After 48 h of incubation, both planktonic cultures and biofilms of *K. schroeteri* (Figure 2C,D) grew better in the presence of the antibiotic compared to after 24 h. The statistically significant inhibition started at a concentration of 6 μg/mL of azithromycin in the case of planktonic cultures (Figure 2C); however, the reaction of *K. schroeteri* biofilms to the antibiotic was peculiar. No significant effect of azithromycin was observed, except that a concentration of 0.001 μg/mL inhibited the biofilm growth by 44%. This correlates with previous data on the inhibition of biofilm growth by low antibiotic concentrations [46].

After 72 h of incubation, an inhibitory effect on planktonic cultures of kytococci was also observed; however, this effect was not so prominent as after 24 h and 48 h of incubation (Figure 2E). Nevertheless, even a weak inhibitory effect (21%) was statistically significant at the concentration of 2 mg/mL of azithromycin in the medium. Inhibition by 51% was observed only at a concentration of 40 μg/mL. In the case of biofilms, a tendency toward growth stimulation was observed; however, no statistically significant differences were found (Figure 2F).

Hence, we decided to use the concentrations of 0.001 μg/mL (as the only concentration significantly affecting the biofilms of *K. schroeteri*) and 4 μg/mL for the following experiments. Both concentrations are subinhibitory.

### 3.2. The Effect of ANP and Azithromycin Combination

After the azithromycin concentrations selection, the next step was to analyze how the addition of ANP may alter the action of the antibiotic. The hormone concentration test was performed previously [19]. Hence, we used 0.65 nM of ANP in the medium because of its effect on the binary biofilms of *S. aureus* and *K. schroeteri* as it was demonstrated. First, we studied how the combinations of azithromycin and the hormone affected the monospecies planktonic cultures and biofilms (Figure 3). As for individual compounds, the effect of compound combinations depended on the cultivation time and azithromycin concentration. This is evidence of complex processes which are triggered by a combination of the compounds. First, both bacteria were mostly affected after 48 h of incubation; thus, this incubation time seem to be optimal in order for both species to obtain biofilms and for researchers to measure the effects of active compounds.

As was shown previously [19], the hormone ANP did not affect the monospecies planktonic cultures and biofilms of both *S. aureus* and *K. schroeteri*; however, the presence of the second counterpart in the community led to the appearance of an inhibitory effect of ANP: *K. schroeteri* in binary community shifted the hormone effect from neutral to inhibitory [19]. Here, in experiments with ANP-azithromycin combinations we found that in *K. schroeteri* planktonic cultures, an addition of ANP increased the inhibitory effect of the antibiotic from 14% to 48% (Figure 3A) after 48 h of incubation. The *K. schroeteri* biofilms were not sensitive to the higher concentration of azithromycin, and the addition of ANP did not change the situation (Figure 3B). However, at the lower concentration, where azithromycin inhibited the growth of the biofilm after 48 h of incubation (Figure 2D), the addition of ANP led to the complete removal of the inhibitory effect of azithromycin (Figure 3B) after 48 h of incubation.

Planktonic cultures of *S. aureus* (Figure 3C) demonstrated controversial behavior in the presence of different azithromycin concentrations. Here, the addition of ANP led to an inhibition of planktonic growth by 15% in the presence of 0.001 μg/mL azithromycin in the medium, while the antibiotic itself did not have any effect. Azithromycin, at the higher concentration, inhibited the planktonic growth by 88%; however, the addition of the hormone decreased the inhibitory effect to 71%. All of these changes occurred after only 48 h of incubation.

Hence, the addition of ANP in the medium may have led to the change in the antibiotic action on monospecies planktonic cultures and biofilms of *S. aureus* and *K. schroeteri*. However, this modulating effect depends on the time of cultivation and the concentration of the antibiotic. This could suggest the multitarget mode of action of both compounds.

### 3.3. Study of Binary Biofilms by CFU Counting

The first assay to primarily evaluate the biomass of each microorganism in the binary biofilm was simple CFU counting. *K. schroeteri* and *S. aureus* form colonies which are easily distinguished from each other; therefore, the binary community of these bacteria is convenient to analyze for the number of CFU. While *K. schroeteri* H01 on the solid RCM forms light beige rough, bulging colonies about 2–3 mm in diameter, *S. aureus* 209P grows in nitidous smooth colonies colored from light yellow to egg yolk color. Also, staphylococcal colonies are normally smaller than kytococcal (1–2 mm in diameter). We additionally studied the amount of CFU in the presence of ANP after 48 h of incubation due to the lack of these data in the previous work [19]. Previously, we found that ANP is able to change the cell surface properties in both bacteria, which resulted in shifts in aggregation and CFU counts after 24 h and 72 h of incubation [19]. Here, first, we also studied biofilms in the system with reduced initial adhesion on the surface of a solid RCM medium. The amount of *K. schroeteri* CFU in monospecies biofilms decreased in the presence of 4 μg/mL of azithromycin in the medium with all incubation durations (Figure 4A), and especially in mature 48 and 72 h biofilms. The addition of the hormone did not change the effect of the antibiotic on the amount of CFUs. In binary biofilms, it was difficult to count the amount of *K. schroeteri* CFUs; in most cases, there were no CFUs on the plates, except the 24 h and 72 h samples with the 4 μg/mL azithromycin concentration. This was the same as we observed previously: no visible CFUs of kytococci was found after forced biofilm dispersion [16]. This all suggests that, potentially, (i) kytococci were at least partially inhibited by staphylococci (potentially due to their lower growth rate), which was partially proved by the azithromycin-mediated staphylococci inhibition, or (ii) the low amount of kytococcal CFU is mediated by aggregation processes, which was checked below. Nevertheless, several points here must be noted. First, a very demonstrative effect of azithromycin in the higher concentration was observed: while *S. aureus* was inhibited, *K. schroeteri* obtained the ability to grow more actively as a more resistant bacterium. Second, the presence of ANP led to a decrease in such a resistance in kytococci in the binary community: no colonies was found on the agar plates when two compounds were combined (Figure 4B). Third, ANP after 48 h of incubation led to very high increase in the *K. schroeteri* CFU count in binary biofilms (Figure 4B), while in monospecies biofilms h the hormone did not affect the kytococcal CFU count. This at the same time contradicts and correlates with results obtained earlier because in monospecies biofilms of *K. schroeteri* ANP increased the amount of CFU after 24 h and 72 h of cultivation [19]; however, such an ambiguity may be explained by the time-dependent effect of the hormone.

In the case of *S. aureus*, in the same manner, azithromycin at a concentration of 4 μg/mL significantly decreased the amount of CFUs in the monospecies biofilms (Figure 4C), while in the control samples, the amount of CFUs was between 1 × 10^9^ and 4 × 10^9^ CFU per biofilm, and in the presence of azithromycin, it decreased to values between 2 × 10^5^ and 2.55 × 10^6^, depending on the cultivation time. In binary biofilms, this value oscillated within similar ranges, and the addition of AMP did not affect the CFU count of staphylococci, nor did it in monospecies or in binary (Figure 4D) biofilms cultivated on the solid medium. In the same time it is interesting that generally the *S. aureus* CFU count was not altered significantly if we compared monospecies and binary biofilms, and this matches with the data obtained previously [19]. The exception are 72 h samples with the addition of 4 µg/mL azithromycin and ANP: in binary biofilms, the amount of staphylococcal CFUs was much lower than in monospecies biofilms (4.7 × 10^5^ and 5.8 × 10^7^). This suggests that (i) kytococci increase the inhibitory effect of azithromycin on staphylococci and (ii) kytococci were not eradicated in binary samples despite the lack of CFUs. This approves the possibility of *K. schroeteri* to provide of at least indirect negative effect on *S. aureus* showed previously [19] and approves the complexity and importance of interactions between species in multispecies biofilms.

As for the second model system in the liquid RCM medium, when the initial adhesion was not reduced in time, the biofilm’s behavior differed significantly, and the effects of the active compounds were also different. First, the amount of CFU per biofilm was generally several times lower (Figure 5) than in the biofilms on solid medium in the first system. Next, the biofilms of kytococci were much more susceptible to azithromycin at the higher concentration (Figure 5A), which is a logical consequence of prolonged cell susceptibility due to the fact that the initial adhesion was not reduced. The addition of ANP led to peculiar results: (i) it increased the inhibitory effect of azithromycin after 24 h of incubation, decreased the inhibitory effect after 48 h, and once again increased the inhibition after 72 h of incubation. This suggests the dependence of ANP action on the biofilm’s formation stage and maturation. Moreover, after 48 h of incubation, the addition of ANP led to a decrease in the CFU amount in the presence of 0.001 μg/mL azithromycin in the medium. This is additional proof that (i) the effect of ANP and the antibiotic is time-dependent; and (ii) 48 h of incubation is an important time point at which to observe the effects of active compounds in the case of the studied bacteria. Also, the lack of *K. schroeteri* CFUs even in presence of 4 µg/mL of azithromycin in binary communities suggests stronger inhibition caused by staphylococci. This is also important because in this system, as it was mentioned above, cells exist longer in more susceptible planktonic form, and the rate of biofilm phenotype formation is crucial for *K. schroeteri* survival. It is interesting that despite ANP stimulating the CFU amount in monospecies *K. schroeteri* biofilms after 24 and 72 h of incubation as it was demonstrated previously [19], after 48 h of incubation, the effect of ANP was inhibitory (Figure 5A): the amount of CFU decreased from 1.9 × 10^6^ to 1.3 × 10^5^. Hence, the effect of ANP is time-dependent.

The number of *S. aureus* (Figure 5B) CFUs was decreased in the presence of 4 μg/mL after all incubation periods. The addition of the hormone led to an increase in the inhibitory effect of azithromycin at higher concentrations after 24 h of incubation. In other samples, there was no effect of compound combinations. In binary biofilms, no CFUs of kytococci were detected; this is potentially a consequence of (i) the competitive interactions between bacteria and (ii) the aggregation processes. The effect of the hormone itself was tested after 48 h of incubation (Figure 5C), and ANP increased the amount of *S. aureus* CFU in binary biofilms from 6.4 × 10^9^ to 1.2 × 10^10^ CFUs per biofilm. No effect of ANP was observed in the samples with antibiotics. However, as it was demonstrated on agar plates (Figure 4), the presence of kytococci in the binary community dramatically affected *S. aureus*. First, the addition of ANP decreased the inhibitory effect of 4 µg/mL azithromycin after 72 h of incubation (Figure 5B). Second, in binary biofilms, this effect of ANP was removed (Figure 5C). Hence, this matches the previously obtained data about *K. schoeteri* as a mediator for *S. aureus*.

Hence, the obtained data match the previous results; the effect of the hormone on kytococci and staphylococci depended on the adhesion, and when the initial adhesion was not reduced, it was able to shift the effect of azithromycin. Also, the second point that matched with the previous data was that the interactions between bacteria is a key factor for the action of active compounds. And these interactions remove the modulating effect of NP on azithromycin’s action against *S. aureus*.

### 3.4. Study of the Metabolic Activity of Monospecies and Binary Biofilms

Despite the pronounced effect of the antibiotic on the number of CFUs, a statistically significant effect of active compounds was observed only on the solid medium (Figure 6). These controversial data suggest that, actually, the amount of cells is not altered, but the aggregation shifts lead to changes in the number of CFUs in biofilms. After 24 h of incubation, the metabolic activity of *K. schroeteri* monospecies biofilms was decreased by 63% (Figure 6A). After 48 h of incubation, 4 μg/mL of azithromycin decreased the metabolic activity by 27%; however, the addition of ANP removed this effect, and the inhibition was only 10% (Figure 6B). Hence, taking into account the number of CFUs, we may propose the aggregation of cells as a key process in biofilms that is affected by active compounds. This also may be proven by the lack of significant differences in other samples (despite the apparent differences in staphylococcal biofilms grown in liquid medium in the presence of higher azithromycin concentrations; see Figure 6E). Also, this may be proven by the OD values in samples with addition of 4 µg/mL azithromycin. The CFU amount of *S. aureus* in binary communities was lower after 72 h of incubation: while the CFU amount decreased in binary communities, the OD of formazan extracts was higher in binary communities. This may suggest on the one hand alterations in adhesion processes in staphylococci, and on the other hand it may be an evidence of presence of kytococci despite the lack of *K. schroeteri* CFUs after binary biofilm dispersion.

### 3.5. Study of Cell Aggregation Strength in Biofilms after Physical Disruption

To understand what exactly changed in the biofilms in the presence of active compounds, we additionally analyzed the aggregation sizes and ratios of the samples. This is an indirect method which allowed us to understand, approximately, the cell volume of each CFU after physical disruption and, thus, how strong the aggregates in biofilms resuspended in the PS are. Representative microphotographs of the samples are presented within the Appendix A.

The first fact important to note is that both in the liquid and on the solid medium, in the absence of the antibiotic, the binary biofilms demonstrated the parameters closer to the monospecies *S. aureus* biofilms (Figure 7 and Figure 8). The aggregation ratio and the aggregate size in staphylococci were generally lower than in kytococci; hence, the binary biofilms had similar meanings. However, in the presence of the higher azithromycin concentration, there was a succession observed when kytococci as a more resistant bacterium turned out to be more presented in the community. That resulted in an increased aggregate size and the aggregation ratio became closer to the values of monospecies kytococcal biofilms. This effect was more prominent after 24 h and 48 h of incubation. In 72 h, the biofilms aggregation of *S. aureus* was higher and the effect was not so visible especially in the liquid RCM.

First, we analyzed the aggregation in biofilms grown on the solid medium, where the initial adhesion stage was reduced in time. First, *K. schroeteri* monospecies aggregates were the largest among the three biofilm types; *S. aureus* produced the smallest ones; and binary biofilms were in between in size. During the cultivation, the size of the aggregates increased, and after 72 h, in the control samples of monospecies kytococcal biofilms, there were, on average, 30 cells per aggregate (Figure 7E). Such an aggregation tendency seems to be characteristic of kytococci, and this is a potential reason for the difficulties which occurred during the CFU counting process. In most cases, we can propose only tendencies, but not statistical differences in aggregates; however, we can carefully say that in monospecies biofilms, azithromycin at a higher concentration decreased the aggregation size both in staphylococci and kytococci. However, in binary biofilms, its effect was the opposite. Also, at the lower concentration, after 72 h of incubation, azithromycin tended to stimulate the aggregate size in staphylococcal monospecies biofilms (Figure 7E). The hormone addition did not sufficiently change the situation, except in the case of 24 h binary biofilms, where the only combination of ANP and 4 μg/mL azithromycin statistically increased the binary aggregate size from, on average, 4.7 to 9.4 cells per aggregate (Figure 7A).

Concerning the aggregation ratio, in monospecies *K. schroeteri* biofilms, the ratio was higher than expected compared to those of the other studied biofilms; however, the highest ratio was observed after 24 h of incubation (71%, Figure 7B). After 48 h, it did not change, but after 72 h, it decreased to 63% (Figure 7F). In monospecies staphylococcal biofilms, the aggregation ratio was at its maximum after 48 h of incubation (56%, Figure 7D). In general, the same tendencies were observed, as seen during the aggregation size calculation: azithromycin at a higher concentration tended to reduce the percentage of aggregates in monospecies staphylococcal biofilms, especially after 48 h of incubation (Figure 7D). In a binary biofilm sample with 4 μg/mL azithromycin and 4 μg/mL azithromycin + ANP, there was a tendency for the aggregation ratio to increase in comparison with the control samples. Furthermore, the addition of ANP did not result in any statistically significant shifts in the aggregation ratio on the solid medium. Hence, the antibiotic decreases the growth and metabolic activity in monospecies biofilms; however, in binary biofilms, bacterial interactions can potentially lead to at least partial resistance to azithromycin, and the addition of ANP can increase these effects.

Biofilms grown in the liquid RCM medium generally maintained similar tendencies (Figure 8). However, the aggregate size was, in general, much smaller than in biofilms on the solid medium. This is a logical consequence of the physiological environment of the biofilm and the non-reduced initial adhesion stage. The maximal average aggregate size in monospecies *S. aureus* biofilms (3.3 cells) was observed after 24 h of incubation, while after 48 and 72 h, the sizes were lower than three cells per aggregate (Figure 8A). In the case of kytococci, the maximal aggregate size was observed after 48 h of incubation—7.1 cells per aggregate. Hence, in systems with “classical” five-step biofilm life cycles, the maturation of *S. aureus* and *K. schroeteri* occurred after 24 h and 48 h, respectively, while on the solid medium, the cells grew in biofilms continuously and did not have the ability to disperse the biofilm. An addition of ANP to the medium with the higher azithromycin concentration led to a statistically significant increase in the aggregate sizes in binary biofilms after 24 h of cultivation (Figure 8B), from 2.8 to 9.5 cells per aggregate. Also, azithromycin at higher concentrations increased the aggregation in binary biofilms; however, the only significant difference was observed after 72 h of incubation (Figure 8F).

Hence, in both systems, azithromycin inhibited the growth of cellular biomass in biofilms, and in binary biofilms, it apparently shifted the balance to the kytococci. This is suggested because of the increase in aggregate size and aggregation ratio. Also, the length of the initial adhesion stage and physiological environment are important for the behavior and growth of biofilm, which is reflected by the appearance of aggregates of maximal size. Despite the statistical insignificance of these differences in most cases, in some samples, ANP has an impact on aggregation and seems to have an addictive effect with azithromycin. Also, based on the aggregation data, the targets of ANP with azithromycin are probably the kytococcal parts of binary biofilms.

### 3.6. Confocal Microscopy of Biofilms

We analyzed, in the same way as we have previously [19], the biomass density and pixel amount in monospecies and binary biofilms. Representative 3D images of the biofilms are presented in the Appendix A, Appendix A. To conduct this, we analyzed the green and red signals from staphylococcal biofilms and binary biofilms. We calculated the ratio in each sample in order to evaluate the approximate green-labeled biomass of staphylococci in binary biofilms and to calculate the species ratio.

Before the results description, it is worth generalizing that only the CLSM allowed us to find the biomass of kytococci in the binary biofilms. In the same way as it was demonstrated earlier, the biomass of kytococci (even calculated indirectly) was about a half of the binary biofilm biomass (Figure 9 and Figure 10). Hence, despite the inability to find *K. schroeteri* by CFU counting, direct visualization confirmed its presence in the community. Hence, the absence of CFU seems to be indeed a result of very strong aggregation. Also, the addition of ANP in some cases altered the mode of action of azithromycin.

First, we analyzed the biomass density of the biofilms (Figure 9). A screening of 24 h biofilms stained with SYTO9 revealed the strong inhibition of *S. aureus* on the glass bottom in the presence of 4 μg/mL azithromycin (from 1.48 to 0.07 μm^3^/μm^2^). The addition of ANP to the medium did not result in alterations to the biomass density; however, the hormone removed the inhibitory effect of 4 μg/mL azithromycin, and the biomass density of monospecies *S. aureus* biofilms returned to 1.1 μm^3^/μm^2^ (Figure 9A). Monospecies 24 h biofilms of *K. schroeteri* were not significantly altered in their biomass density in the presence of active compounds. Binary 24 h biofilms showed a behavior similar to monospecies staphylococcal biofilms, with some specificities: inhibition in the presence of μg/mL azithromycin was not as pronounced as it was in monospecies *S. aureus* biofilms (from 2.2 to 1.0 μm^3^/μm^2^), which is a logical consequence of the presence of kytococci. However, the addition of ANP to the system first led to significant inhibition of the growth in comparison with the control samples (which is the same as the result described earlier by Diuvenji et al., 2022 [19]): the density decreased from 2.2 to 1.0 μm^3^/μm^2^ (the same as with azithromycin at the higher concentration). And the combination of ANP and 4 μg/mL azithromycin led to a full removal of this inhibitory effect and to an increase in biomass density up to 3.11 μm^3^/μm^2^ (1.5 times higher than in the control samples).

To understand the alterations in the biomass balance in binary biofilms, we stained *S. aureus* using the specific FISH probe. First, we analyzed the red signal from 24 h *S. aureus* monospecies biofilms (Figure 9B) and noticed a phenomenon that, when we optimized the procedure for the new confocal microscope which was used, the red signal was even higher than the green signal from SYTO9. For instance, in the control samples, the red-labeled biomass density was 3.48 μm^3^/μm^2^ instead of 2.2 μm^3^/μm^2^ with green labeling. Thus, for staphylococcal biofilms, we had to calculate the signal ratio at every single point and then re-calculate the approximate green part of the staphylococci as a component of the total green signal of binary biofilms (Figure 9C). But in the beginning, we found that in monospecies biofilms, the red labeling revealed no effect of ANP on biofilms (Figure 9B). This matches fully with previous data [19]. At the same time, there was an inhibitory effect of azithromycin detected at the higher concentration (from 3.48 to 0.8 μm^3^/μm^2^). Hence, there are two potential reasons for this difference (or, likely, their combination). The first is that ANP and azithromycin affect a part of the biofilm which is not appropriate for specific FISH labeling, but is highly susceptible to unspecific SYTO9 staining of the biofilm matrix. Such a part of a biofilm may be only biofilm matrix. The second reason may be a constrained use of imaging optimization, which leads to this controversial result. However, we suggest that these reasons were actually combined, and that the contribution of the first was dominative, because all of the samples were processed in strict accordance with the protocol before blind analysis was performed. Also, the FISH probe should be more specific to the cellular DNA than the SYTO9, which is not fully specific and actually may bind more targets. In the 24 h binary biofilms, at the same time, the red-labeled biomass density of staphylococci decreased in the presence of 4 μg/mL azithromycin (from 2.33 to 0.61 μm^3^/μm^2^). The combination of ANP and azithromycin at the higher concentration led to an increase in the inhibitory effect of azithromycin (which reflected in decrease in the biomass density to 0.09 μm^3^/μm^2^).

When we calculated the approximate biomass ratio in binary biofilms based on the green signal, we found that *S. aureus* decreased in the presence of 0.001 and 4 μg/mL of azithromycin, from 1.57 to 0.56 and 0.64 μm^3^/μm^2^, respectively (Figure 9C). However, neither ANP nor combinations of the hormone and the antibiotic had any effect (except the removal of the difference’s significance). Hence, hypothetically, there could potentially be a substitution of *S. aureus* cellular biomass by the biofilm matrix, the errors in the coefficient calculations, or (most probably) a combination of these factors. Thus, we can only infer that the cellular biomass of *S. aureus* in the binary community decreased in the presence of the combination of compounds, and potentially, this can be at least partly compensated by the increased synthesis of the matrix. “Kytococcal”, part of the binary biofilm after 24 h of incubation, decreased in the presence of 4 μg/mL of azithromycin from 1.43 to 0.74 μm^3^/μm^2^, respectively. ANP also decreased the biomass density to 0.74 μm^3^/μm^2^. Azithromycin, at a concentration of 0.001 μg/mL, did not affect the “kytococcal” part of binary biofilms; however, the combination with ANP led to a decrease in the biomass density to 0.74 μm^3^/μm^2^. Hence, ANP “overlapped” the antibiotic at the lower concentration. However, the combination of ANP with 4 μg/mL of azithromycin led to a full removal of any inhibition of the “kytococcal” portion, and even to a slight stimulation (increase in the biomass density to 2.21 μm^3^/μm^2^). This is interesting because the monospecies biofilms of *K. schroeteri* were not significantly susceptible to any compound or combination. Hence, as was the earlier result, the key role was played by the bacterial interaction in the tightly packed biofilm community.

After 48 h of incubation, in general, azithromycin at the higher concentration maintained the ability to inhibit the biofilm growth of staphylococci labeled with both green SYTO9 and R6G FISH probes (Figure 9D,E). However, the paradoxically statistically insignificant difference between the control and 4 μg/mL azithromycin samples of *S. aureus* monospecies biofilms was higher (0.76 and 0.38 μm^3^/μm^2^) than the statistically significant difference between the 4 μg/mL azithromycin and 4 μg/mL azithromycin + ANP samples (0.38 and 0.14 μm^3^/μm^2^). Kytococcal biofilms and green-labeled binary biofilms were not susceptible to active compounds. Red labeling of the *S. aureus* monospecies and binary biofilms revealed the inhibitory effect of 4 µg/mL azithromycin (Figure 9E). The calculation of an approximate biomass ratio in the binary biofilms revealed an increase in the kytococcal biomass in the presence of a combination of 0.001 µg/mL azithromycin and ANP in comparison with 0.001 µg/mL azithromycin samples (from 0.37 to 2.31 µm^3^/µm^2^, Figure 9F).

Additionally, we studied the pixel volumes of the biofilms (Figure 10). Despite the fact that this approach was less accurate in comparison to the automatic biomass density calculation, some experimental patterns obtained previously were approved. First, staphylococcal biomass in monospecies biofilms decreased in the presence of 4 µg/mL azithromycin in most cases (Figure 10A,B,D,E). Second, ANP in combination with 4 µg/mL azithromycin strongly decreased the amount of *S. aureus* green pixels (i.e., biomass) after 48 h of incubation in comparison with 4 µg/mL azithromycin samples (from 793,573 to 300,315 pixels per visual field; Figure 10D). Also, in 48 h binary biofilms, *S. aureus* was stimulated by 0.001 µg/mL azithromycin from 1.8 × 10^6^ to 4.2 × 10^6^ (Figure 10F). Also, as with the biomass density, the kytococcal biomass was stimulated by the addition of ANP in the presence of 0.001 µg/mL of azithromycin (from 1.99 × 10^6^ pixels to 4.85 × 10^6^ pixels, Figure 10F).

Hence, despite the dependence of the model system, the surface type, and other factors, it was established that ANP is able not only to modify the azithromycin activity against the biofilms of the studied microorganisms, but this combined action also affects the biomass balance inside the community. Also, it was found that ANP is able to shift the balance between microorganisms in the community in the presence of the antibiotic.

### 3.7. Study of Growth Kinetics

As we described in our previous works with other non-motile bacteria, the use of two types of model systems allows us to compare the effects of active compounds on the growth of most biofilms (with forced adhesion) and of most planktonic suspensions (without forced adhesion). All of the calculated values are presented in the Appendix A, Appendix A. Growth curves are plotted in Figure 11. To summarize the results, we can state that azithromycin in both concentrations decreased the growth rate in the monospecies cultures of both microorganisms in each system, while the addition of ANP partly removed this effect. However, in binary biofilms, no such effects were observed (Appendix A). Concerning the growth curves, their shape partly confirmed the previous finding (on the PTFE cubes and in the glass-bottom CLSM plates). Here, in the system with forced adhesion (mostly biofilms), the combination of ANP and 4 µg/mL of azithromycin led to an increase in the *S. aureus* OD after 48 h (Figure 11D), which corresponds to the results for the Teflon cubes (Figure 3). At the same time, in the second model system, the addition of ANP led to a decrease in the *S. aureus* culture OD after 48 h, which corresponds to the results of the CLSM (Figure 9D and Figure 10D). However, this is, rather, “as near as a toucher” because of the impossibility of dividing the biofilm and the planktonic growth. Also, the values of OD in *K. schroeteri* monospecies cultures and in binary communities were quite similar both in samples with the antibiotic and the antibiotic–hormone combinations. Despite the presence of some these uncertainties, there was a strongly exhibited effect of the antibiotic (especially in the higher concentration). Also, we can establish that, as was shown in the previous work [19], the binary communities behaved as a “hybrid” of the two monospecies cultures in both systems (Figure 11C,F). Also, without the forced initial adhesion, the curves were more flattened and the final OD values were generally lower than in the samples with forced adhesion. Hence, the biofilm formation in these non-motile bacteria contributed more to the final OD values.

### 3.8. Gene Expression Study

To understand what potential molecular mechanisms underlie the shifts in the responses to the antibiotic caused by ANP, we investigated the differential gene expression using the qPCR method. The results revealed the ability of all components to affect the expression of genes of resistance to different antibiotics. However, this effect was not directly dependent on the concentration of the antibiotic, and in most cases, only a combination of compounds had an effect on gene expression.

Due to the lack of a complete genome sequence of *K. schroeteri*, it was impossible to design the correct primers directly for this species. Hence, we decided to rely on the complete genome of *Kytococcus sedentarius*—the species closer to *K. schroeteri*. Hence, in kytococci, we found two genes with macrolide resistance (Table 2). The first was the gene *abeS* (38.14% of identity) of the drug efflux pump of the small multidrug resistance (SMR) family [47]. This gene was stimulated in expression only in the presence of the combination of ANP and azithromycin at the higher concentration. Similar results were obtained for another macrolide resistance gene, *mrx* (27.25% identity). This gene encodes a putative transmembrane protein; however, its functions are still unknown [48,49], and we could suggest that this protein can also be an efflux pump. At the same time, the *oleC* gene (30.74% of identity) expression was affected by neither the hormone nor the antibiotic or compound combinations. Nevertheless, we suggest that, potentially, the mechanism of ANP’s action lies in increasing the specific macrolide efflux activity in kytococcal cells in the presence of antibiotics at higher concentrations.

It is worth noting that not only genes of macrolide resistance were altered in expression in the presence of active compounds. The gene of *K. schroeteri* closer to *arlR* (35.62% of identity)—a regulatory gene of the two-component ArlR-ArlS system of *S. aureus* which participates in the regulation of fluoroquinolone resistance [50]—was downregulated in the presence of all compounds except the combination of ANP and 4 µg/mL azithromycine in the medium. Interestingly, this gene is also connected with efflux pump regulation. Another gene, *smeS* (30.31% of identity), a kinase component of the two-component regulatory system SmeR-SmeS, which is responsible for resistance to aminoglycoside antibiotics, cephalosporins, cephamycins, and penams, and regulates efflux pumps [51], was downregulated by ANP. A similar picture was observed with *tetA* (37.55% of similarity), a gene of a tetracycline efflux pump [52]. Finally, another gene of fluoroquinolone resistance, *mdtK* (26.78% of similarity), was downregulated in the presence of 0.001 µg/mL of azithromycin in the medium. This gene encodes an efflux pump for antibiotic removal from cells [53]. Other genes (Table 2) were not altered in their expression.

Thus, macrolide resistance was stimulated in kytococci only in the presence of ANP and only in the presence of higher amounts of azithromycin. At the same time, the resistance to other antibiotics was potentially reduced. Also, both the hormone and the antibiotic affected efflux pumps; thus, cell transport was the potential target of these active compounds in kytococci.

In staphylococci, no genes of macrolide resistance were found (Table 3). However, other resistance genes were upregulated by the combination of 0.001 µg/mL azithromycin and ANP: *mgrA*, a regulator of multidrug efflux pumps [54]; *norC*, a fluoroquinolone efflux pump [55] regulated by MgrA; and *vanT_G_*, a gene of glycopeptides (especially vancomycin), and membrane-associated racemase [56]. *mgrA* was also upregulated by the combination of 4 µg/mL azithromycin and ANP. Hence, it is possible that the bacterium was more resistant to other antibiotics in the presence of the antibiotic–hormone combinations (this is the opposite effect in comparison to kytococci). In addition, except for *vanTg*, all of the genes encoded efflux pumps, and all of the genes encoded membrane-associated proteins.

## 4. Discussion

The data obtained reveal, once again, the complexity of the interactions between microorganisms inside the microbial community and between the human organism and its microbiota. First, interactions between staphylococci and kytococci lead to the alterations of effect of azithromycin and ANP. Second, ANP is able to alter the action of azithromycin on bacterial biofilms, especially in binary communities. This effect depends on the cultivation system and the presence of a liquid phase, as well as on the surface properties and incubation time. Here, on the hydrophobic PTFE cubes, ANP attenuated the inhibitory effect of azithromycin, while on hydrophilic glass, the hormone strengthened the antibiotic. The study of cell aggregation revealed the effect of the combination of azithromycin and ANP on cell coaggregation. Hence, all of the data show that the effect of ANP (and a combination of ANP and azithromycin) is targeted on cell surface properties and aggregation, which confirms the data obtained previously [19]. The key points are, first, that the most pronounced effect was observed in the presence of a combination of azithromycin (especially at higher concentrations) and the hormone. This effect was observed in all systems, and it is interesting that this combination was able to decrease the staphylococcal biomass in the binary communities, which was observed using the CLSM. And the fact that a combination of the hormone and the antibiotic also resulted in differential expression levels of antibiotic resistance genes is worth noticing. This is especially interesting not only because of macrolide resistance genes being altered in their expression, but also due to genes with resistance to a rather wide range of antibiotics. These data suggest that, in the case of improper antibiotic administration that leads to lower drug concentrations in an organism, the reaction of the bacterial community may, in fact, be unpredictable. And in some cases, ANP, as a hormone which is present at higher concentrations in blood plasma in humans with different types of heart failure [57,58], may play the role of antibiotic attenuator. This provides a suggestion of the potential inefficacy of azithromycin treatment for infections in patients with heart failure. Another example would be, for instance, that improper antibiotic combinations of azithromycin with other drugs may not be effective due to a decrease in resistance in the presence of an azithromycin–ANP combination. However, of course, such suggestions are rather ephemeral and need to be proven by robust experimental studies. The second key point is that bacterial interactions inside multispecies biofilms, as it was demonstrated previously [19,20] may lead to the critical alterations in the effects of active compounds such as antibiotics or hormones or any others. Thus, as *K. schroeteri* seems to be much less dangerous than *S. aureus* and hence may be considered as a “true” commensal bacterium, it can modify the effect of both ANP and azithromycin, making *S. aureus* more sensitive to those compounds. And that is an additional evidence of a protective function of skin commensal microbiota.

The investigation of interactions between human organisms and its microbiota in recent decades have transitioned from a rather rare and exotic research subject to an intensively studied area. However, as we discover more, more questions and blank spots arise. The effects of human humoral factors on commensal (and pathogenic) microbiota of the human gut and skin are becoming rather obvious, despite the tiny quantity of scattered data. And if we dive into the question of how humoral factors influence the activity of other compounds on bacteria, such as antibiotics, we find ourselves in unknown territory. The problem becomes even more difficult to be solved when it interferes with another almost-unexplored area of interspecies interactions inside the microbial communities. Lots of studies (especially last decades) have been conducted into the microbial communities in different habitats, and lots of data have been collected about the species and genes contains using omics techniques; however, interspecies relations seem to remain truly “terra incognita”. The presence of a second counterpart in the simplest multispecies system—binary biofilm—may alter significantly not only the behavior of microorganisms, but also modify the manner of action of active compounds. If we also take into account the model system conditions, the physico-chemical properties, time, temperature, etc., we will end up with indeed an endless field of research. Nevertheless, these areas seems to be an extremely important to study. The problem of bacterial resistance to antibiotics has long been “the talk of the town”, not only among medical microbiologists, but also among everyone who is somehow involved in issues related to the treatment of bacterial infections. The annual increase in multidrug-resistant strains and nosocomial infections cases, the decrease in antibiotic efficacy, the ability of pathogens to “hide away” inside dense biofilm communities and the high cost of new drug development seem to be leading mankind to the “red line” representing the end of a comfortable antibiotic era. And the here-obtained data constitute a small step towards understanding the complexity of natural processes in human bodies and human–microbiota and microbial interspecies interactions. And we hope that someday, in the future, our data will help in the development of new drugs or strategies for the treatment of bacterial infections, especially taking into account the complex interactions of microorganisms in biofilms and the potential influence of humoral regulatory factors on them.

## 5. Conclusions

It was demonstrated once again that interspecies relations inside multispecies biofilms play a crucial role in bacterial behavior and alter the reactions of bacteria on active compounds. Here, the presence of *K. schroeteri* increased the inhibitory effect of the ANP and azithromycin combination on *S. aureus*. Concerning the active compounds, it was established that ANP, on the one hand, is able to shift the balance between bacteria in binary biofilms of *K. schroeteri* and *S. aureus*, and on the other hand, the hormone alters the effect of azithromycin in subinhibitory concentrations on the bacterial biofilms. The effects of compounds seem to be related mostly to the surface properties of cells, but they also reflect the differential expression of antibiotic resistance genes. Most of the genes altered in expression belong to the efflux pump class. Also, the combination of azithromycin and ANP can potentially affect matrix synthesis in biofilms. Hence, we found that the mechanisms of action of these active compounds are very complex and difficult to solve. However, this is a promising perspective for future research, as it demonstrates the potential to investigate hormones as adjuvants for antibiotics and to provide new strategies for the medical treatment of different infections.

## Figures and Tables

**Figure 1 microorganisms-11-02965-f001:**
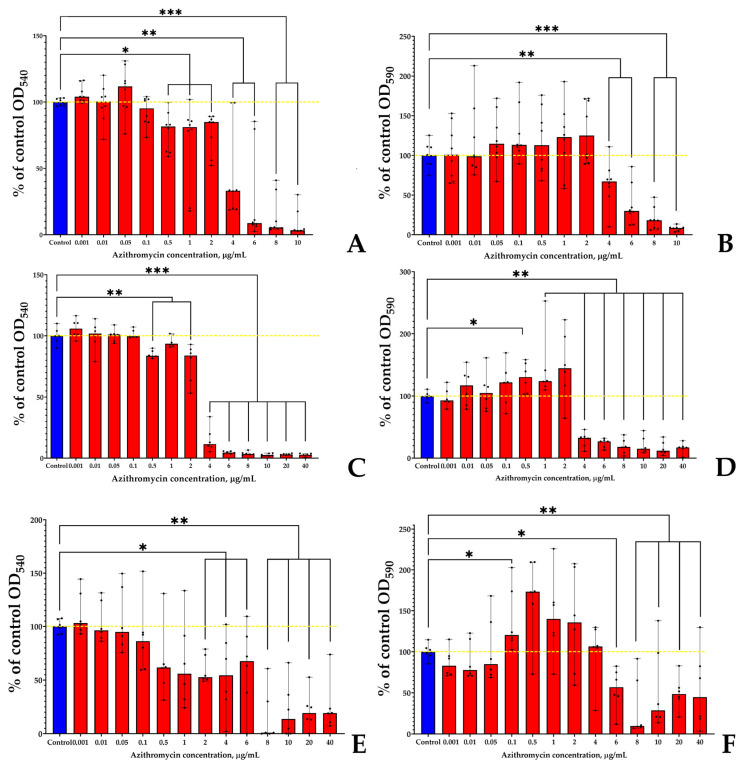
Effects of different azithromycin concentrations on *S. aureus* planktonic cultures (**A**,**C**,**E**) and biofilms on the PTFE cubes (**B**,**D**,**F**) after 24 h (**A**,**B**), 48 h (**C**,**D**), and 72 h (**E**,**F**) of cultivation. Yellow dotted lines mean 100% level; *: *p* < 0.05; **: *p* < 0.01; ***: *p* < 0.001.

**Figure 2 microorganisms-11-02965-f002:**
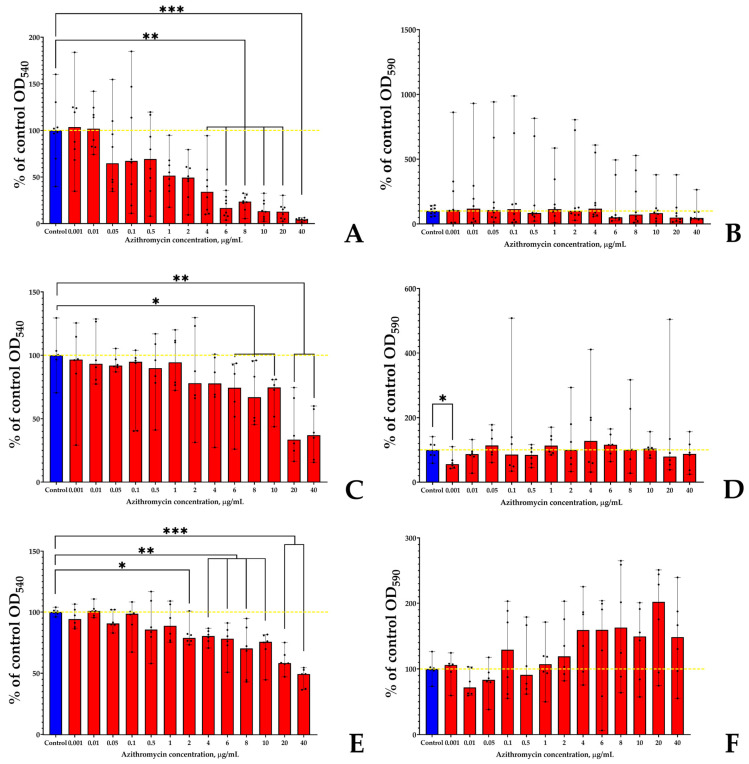
Effects of different azithromycin concentrations on *K. schroeteri* planktonic cultures (**A**,**C**,**E**), and biofilms on the PTFE cubes (**B**,**D**,**F**), after 24 h (**A**,**B**), 48 h (**C**,**D**), and 72 h (**E**,**F**) of cultivation. Yellow dotted lines mean 100% level; *: *p* < 0.05; **: *p* < 0.01; ***: *p* < 0.001.

**Figure 3 microorganisms-11-02965-f003:**
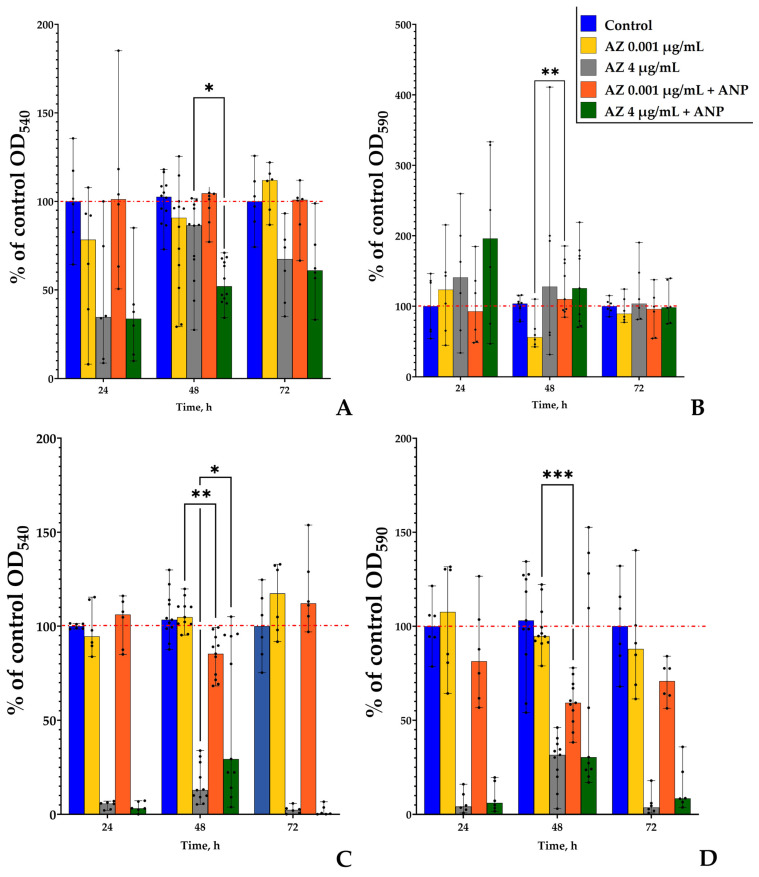
Effect of combinations of azithromycin and ANP on planktonic cultures (**A**,**C**) and biofilms (**B**,**D**) on the PTFE cubes of *K. schroeteri* (**A**,**B**) and *S. aureus* (**C**,**D**). Red dotted lines mean 100% levels. *: *p* < 0.05; **: *p* < 0.01; ***: *p* < 0.001.

**Figure 4 microorganisms-11-02965-f004:**
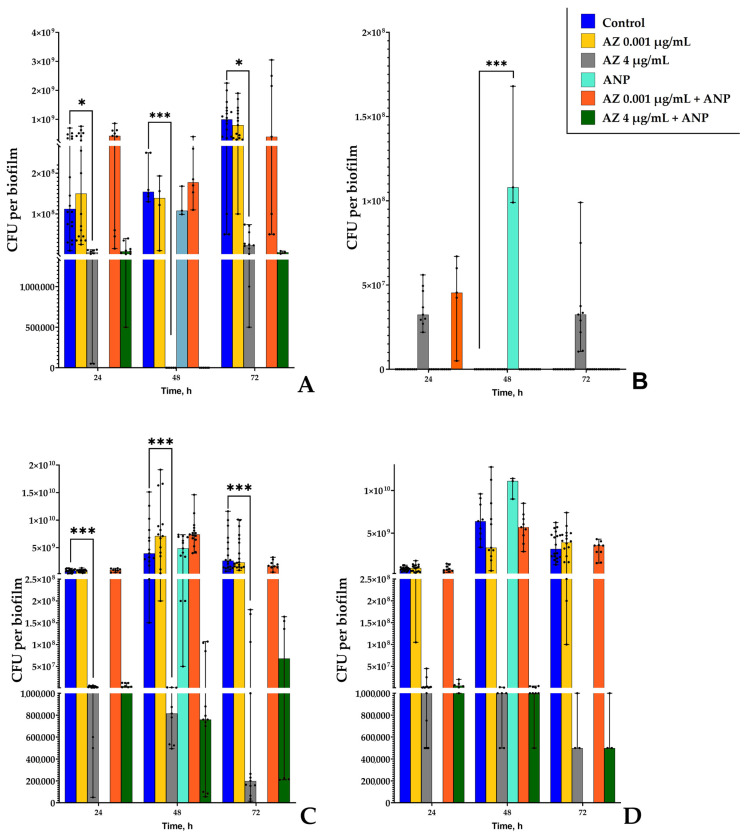
CFU counts in biofilms cultivated on the GMFF on the solid RCM medium, in the presence of active compounds and their combinations. (**A**) Number of CFUs of *K. schroeteri* in monospecies biofilms; (**B**) number of CFUs of *K. schroeteri* in binary biofilms; (**C**) number of CFUs of *S. aureus* in monospecies biofilms; and (**D**) number of CFUs of *S. aureus* in binary biofilms. *: *p* < 0.05; ***: *p* < 0.001.

**Figure 5 microorganisms-11-02965-f005:**
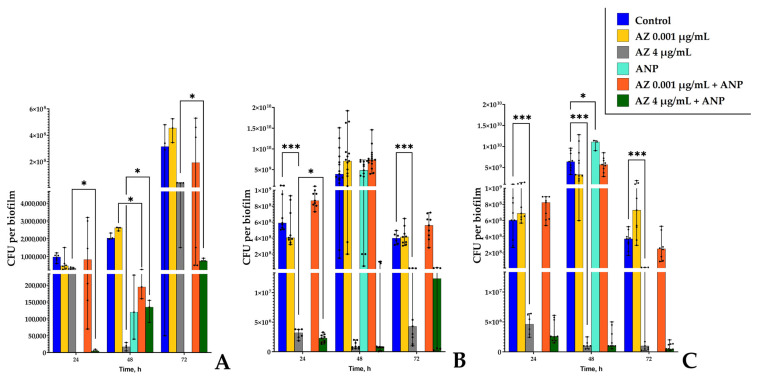
CFU counts in biofilms cultivated on the GMFF in liquid RCM medium in the presence of active compounds and their combinations. (**A**) Number of CFUs of *K. schroeteri* in monospecies biofilms; (**B**) number of CFUs of *S. aureus* in monospecies biofilms; and (**C**) number of CFUs of *S. aureus* in binary biofilms. *: *p* < 0.05; ***: *p* < 0.001.

**Figure 6 microorganisms-11-02965-f006:**
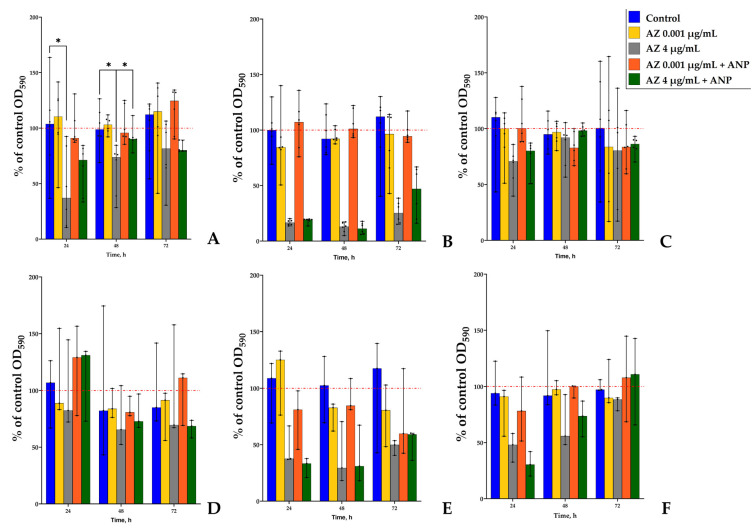
The MTT staining of biofilms cultivated on the solid (**A**–**C**) and in the liquid (**D**–**F**) RCM medium in the presence of active compounds and their combinations. (**A**,**D**): monospecies *K. schroeteri* biofilms; (**B**,**E**): monospecies *S. aureus* biofilms; (**C**,**F**): binary biofilms. *: *p* < 0.05.

**Figure 7 microorganisms-11-02965-f007:**
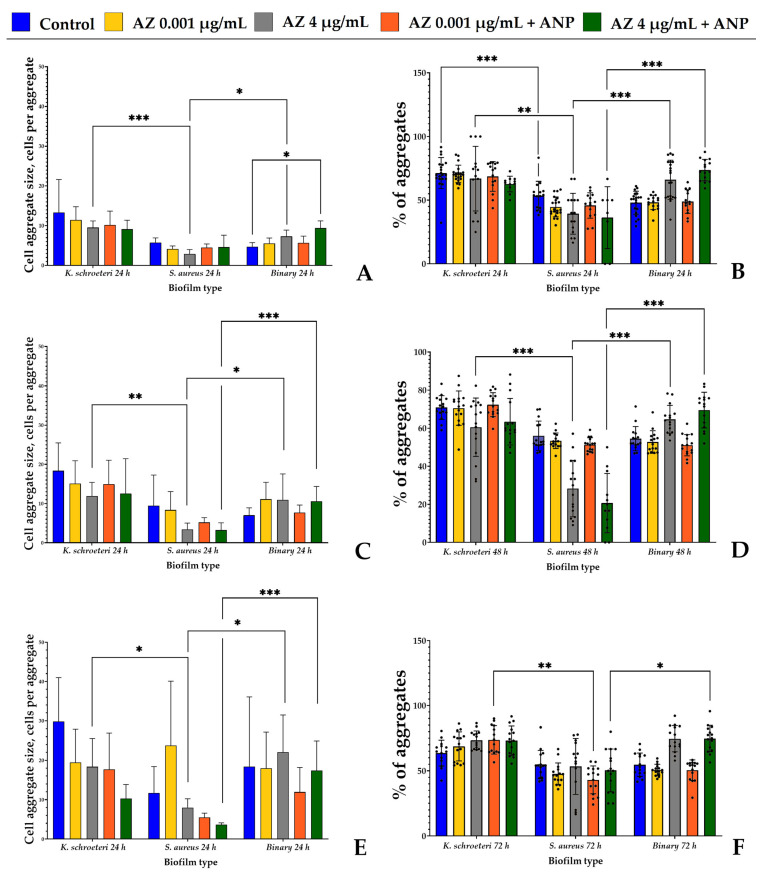
Cell aggregate size (**A**,**C**,**E**) and percentage of aggregates (**B**,**D**,**F**) in monospecies biofilms of *K. schroeteri* (**A**,**B**), *S. aureus* (**C**,**D**), and binary biofilms (**E**,**F**) cultivated on the solid RCM medium in the presence of active compounds and their combinations. *: *p* < 0.05; **: *p* < 0.01; ***: *p* < 0.001.

**Figure 8 microorganisms-11-02965-f008:**
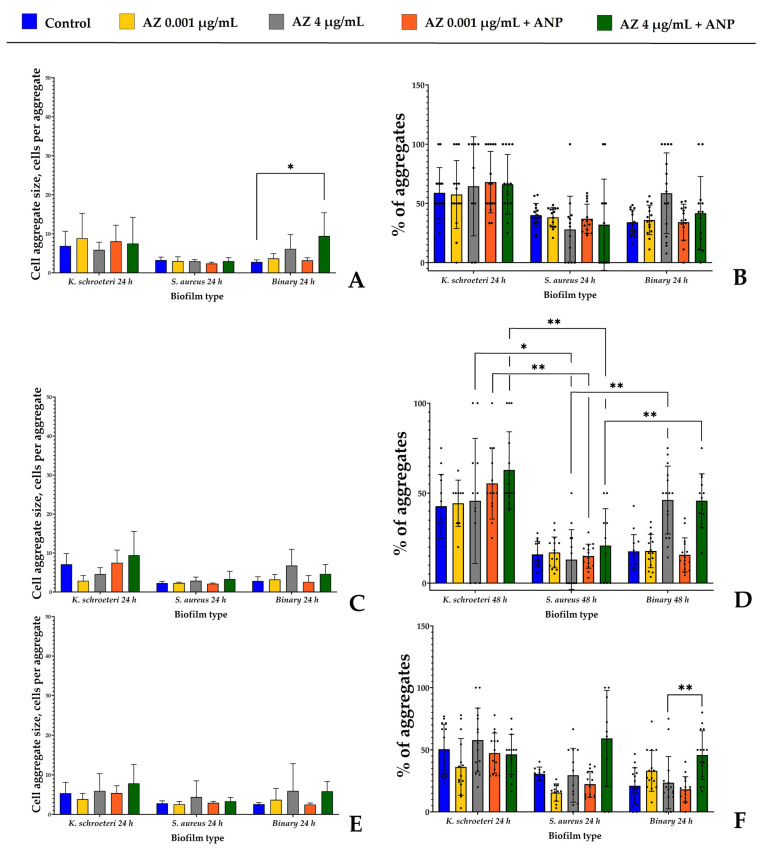
Cell aggregate size (**A**,**C**,**E**) and percentage of aggregates (**B**,**D**,**F**) in monospecies biofilms of *K. schroeteri* (**A**,**B**), *S. aureus* (**C**,**D**), and binary biofilms (**E**,**F**) cultivated in liquid RCM medium in the presence of active compounds and their combinations. *: *p* < 0.05; **: *p* < 0.01.

**Figure 9 microorganisms-11-02965-f009:**
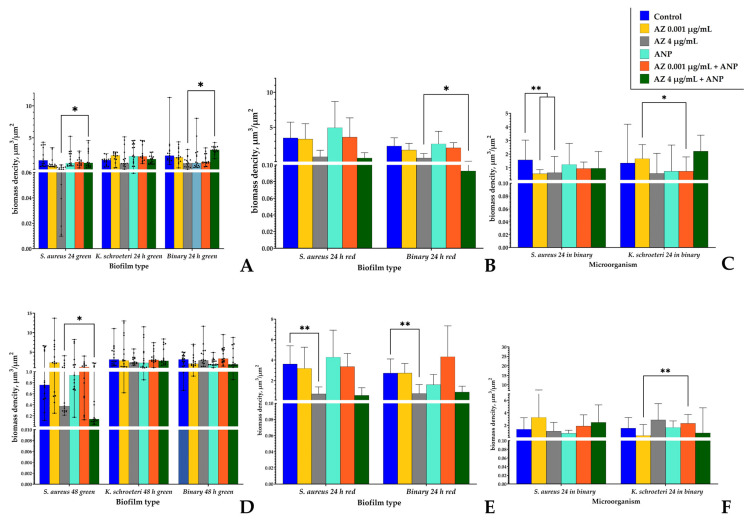
Processing CLSM images. Biomass density of monospecies and binary biofilms of *K. schroeteri* and *S. aureus* stained with SYTO9 Green (**A**,**D**); monospecies and binary biofilms where *S. aureus* was labeled with R6G using FISH (**B**,**E**); and ratio of *S. aureus* and *K. schroeteri* in binary biofilms (**C**,**F**). *: *p* < 0.05; **: *p* < 0.01.

**Figure 10 microorganisms-11-02965-f010:**
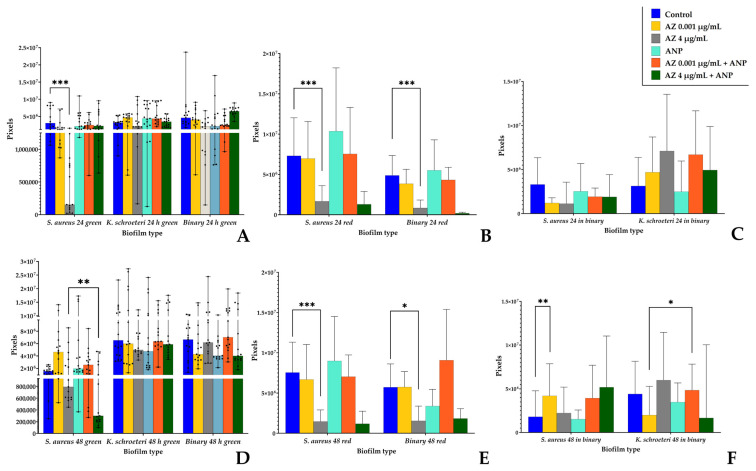
CLSM image processing. The total pixel volume of the monospecies and binary biofilms of *K. schroeteri* and *S. aureus* stained with SYTO9 green (**A**,**D**); monospecies and binary biofilms where *S. aureus* was labeled with R6G using FISH (**B**,**E**); ratio of *S. aureus* to *K. schroeteri* in binary biofilms (**C**,**F**). *: *p* < 0.05; **: *p* < 0.01; ***: *p* < 0.001.

**Figure 11 microorganisms-11-02965-f011:**
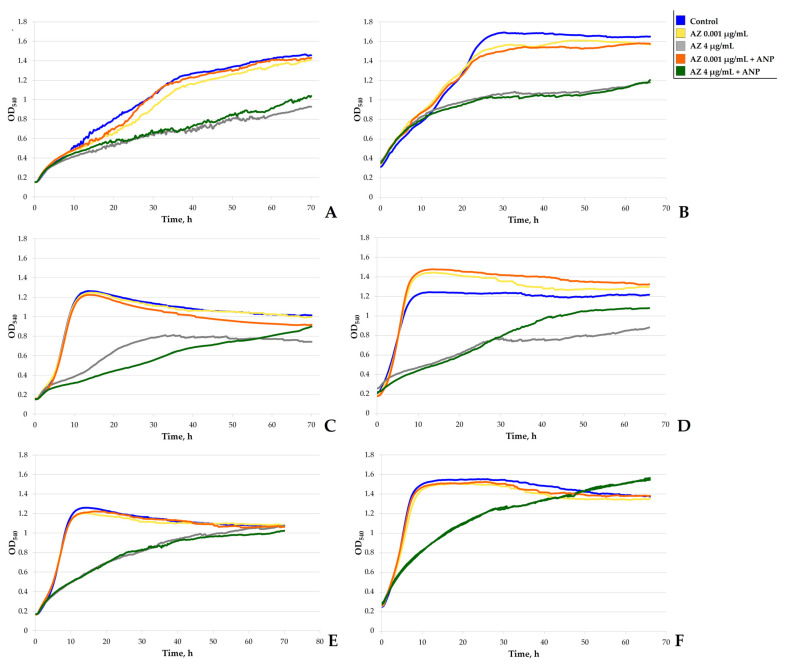
Kinetic curves of monospecies (**A**–**D**) of *K. schroeteri* (**A**,**B**), *S. aureus* (**C**,**D**), and binary (**E**,**F**) cultures grown in the system without the forced initial adhesion step (**A**,**C**,**E**) and in the system with forced initial adhesion (**B**,**D**,**F**).

**Table 1 microorganisms-11-02965-t001:** Primers used for qPCR.

Gene	Primer	Sequence 5′–3′
*K. schrtoeteri*		
FomB (phosphonic acid antibiotic)	Forvard	CCTGGTGAAGGGTGATCCTG
	Reverse	CGTAGACGTGCGTAGAGGAG
MecA (meropenem)	Forvard	GGAGATGATGGTGTCCGTGG
	Reverse	CTCTTCCATCATCGAGCGGG
AbeS (macrolides, aminocoumarins)	Forvard	CCGGATTCACCAACCCTCTC
	Reverse	AACCCCTTGATCAGCGTCAG
OleC (macrolides)	Forvard	CAGAGTTGATGGCCCTGGAG
	Reverse	CCGCTCGTAGAAGGGAATCG
NovA (aminocoumarins)	Forvard	GAACTTCCTGCCCTCGTTGG
	Reverse	CCGAGATGGTCGAGAGAAGC
Mrx (macrolides)	Forvard	CTGGTGGTCTTCAAGGTGCT
	Reverse	CGAGGAAGCCGATGACGTAG
MdtK (fluoroquinolones)	Forvard	TGTTCGTCTTCCTCGCCTAC
	Reverse	GATGCGTAGGTAGGTGACCC
ArlR (fluoroquinolones)	Forvard	GTGAACCAGACCCAGACCG
	Reverse	GAAGGGCTTGGTGACGTAGT
BaeR (aminoglycosides, fluoroquinolones)	Forvard	GGAGGGTGACAGCATCGAC
	Reverse	GAAGGTCAGGTCCAGCGTG
SmeS (aminoglycozides, cephalosporines)	Forvard	ATCCAGAGTCCGATCAACGC
	Reverse	TCACGGTGGATAAACGGGTG
TetA (tetracyclines)	Forvard	CGTCCCAGCAAGACCTACTC
	Reverse	GCTCGTCCAGGAAGATGACC
16 s rRNA	Forvard	TCAACCGTGGAGGGTCATTG
	Reverse	TGCACCACCTGTCACTTTGT
*S. aureus*		
MepR (tetracyclines)	Forvard	GCATTACAACGAACAGGTCCA
	Reverse	TCCCAGAGGTAGTCAGCCC
MgrA (peptide antibiotics, tetracyclines, cephalosporines)	Forvard	AGCGTGAACGTTCCGAAGTC
	Reverse	GAAGCTGAAGCGACTTTGTCA
NorC (fluoroquinolones)	Forvard	TTGTTGTTGGAGCAGGTGGT
	Reverse	CAGGCGTCCCTTTGATGAGT
VanTG (glycopeptides)	Forvard	CTTTGCCTGTGCTGACGAAC
	Reverse	ACCTCTACCGACTGTGGACT
LmrS (oxazolidine, macrolides, phenicols, aminoglycosides)	Forvard	TGGACCTGCGCTGCTTATAC
	Reverse	AGCCGTGCCATGTGAGATTT
SdrM (fluoroquinolones)	Forvard	TGGGCATAGTTGGCAGTGTT
	Reverse	ATGGCAATGATCGCAATCGG
16 s rRNA	Forvard	TCAACCGTGGAGGGTCATTG
	Reverse	TGCACCACCTGTCACTTTGT

**Table 2 microorganisms-11-02965-t002:** Differential expression of genes of antibiotic resistance in *K. schroeteri*, measured by qPCR.

Gene	Amount of Amplification Product in Relation to Control Sample, 2^−ΔΔCt^				
	ANP	Az 0.001 µg/mL	Az 4 µg/mL	Az 0.001 µg/mL + ANP	Az 4 µg/mL + ANP
*fomB* (phosphonic acid antibiotic)	2.1	1.9	−1.9	0	1.8
	0	−97.6	23.3	−6.9	−1.9
	1.4	−1.7	−1.4	1.7	−64
*mecA* (meropenem)	−4.5	−2.7	3.1	−6.6	1
	3.7	1.6	−1.2	19	4
	12.5	−1.4	1.2	1.9	−1.4
*abeS* (macrolides, aminocoumarins)	4.6	0	0	0	** 6.7 **
	−3.8	−76.1	−19.2	−7.2	** 5.4 **
	1.2	2.8	2	4.2	** 2 **
*oleC* (macrolides)	−4.9	−2.9	0	−4.4	−2.5
	1.4	−29.4	−5.9	−1.2	−2.4
	0	2.1	1.3	3	12.2
*novA* (aminocoumarins)	1.7	0	3.8	0	2.5
	1.7	1.1	−1.6	6.8	10.9
	0	−6.3	2.5	−3.6	−3.5
*mrx* (macrolides)	−5.5	1.1	−2	−1.2	** 6.7 **
	1.3	−2	1.8	3.1	** 1.6 **
	1.1	2.5	1.3	1.8	** 1.6 **
*mdtK* (fluoroquinolones)	−1	** −3.4 **	−1.1	−2.7	2.8
	1.7	** −1.7 **	1.4	−1.9	−1.5
	1.1	** −1.7 **	1.8	2.3	−6.6
*arlR* (fluoroquinolones)	** −2.2 **	** −4.8 **	** −1.3 **	** −1.6 **	−3.5
	** −1.1 **	** −6.8 **	** −2.8 **	** −1.8 **	−2.4
	** −6.3 **	** −8.5 **	** −1.2 **	** −2.9 **	7.8
*baeR* (aminoglycosides, fluoroquinolones)	** −5.3 **	** 1.8 **	** −1.7 **	** −1.7 **	−1.3
	** −5.3 **	−4.3	−4.3	1.6	1.6
	** −2 **	1.7	1.6	−1.5	16.2
*smeS* (aminoglycozides, cephalosporines)	** −11.1 **	−5.4	−1.6	−1.6	33.4
	** −11.1 **	5	−5.4	2	−1.1
	** −2.1 **	1.9	2	−1	11.1
*tetA* (tetracyclines)	** −1.9 **	−4.5	−5.2	−1.3	0
	** −1.4 **	2.1	0	15	15.6
	** −1.3 **	5.3	2.4	0	−3.9

Green font indicates statistical reproducibility of the results.

**Table 3 microorganisms-11-02965-t003:** Differential expression of genes with antibiotic resistance in *S. aureus* measured by qPCR.

Gene	Amount of Amplification Product in Relation to Control Sample, 2^−ΔΔCt^				
	ANP	Az 0.001 µg/mL	Az 4 µg/mL	Az 0.001 µg/mL + ANP	Az 4 µg/mL + ANP
*mepR* (tetracyclines)	−1.25	−11.55	−1.1	−1.67	2.45
	−45.25	2.91	−11.96	5.71	−1.82
	1.37	1.1	3.32	1.59	1.79
*mgrA* (peptide antibiotics, tetracyclines, cephalosporines)	1.42	2.25	1	** 1.51 **	** 1.91 **
	−4.56	−2.89	−5.82	** 8.44 **	** 1.83 **
	1.02	−1.14	2.87	** 3.51 **	** 1.78 **
*norC* (fluoroquinolones)	4.17	20.25	4.47	** 3.81 **	4.17
	−3.46	−2.06	−2.81	** 1.27 **	−3.71
	−2.48	1.53	2.08	** 8.75 **	−1.17
*vanT_G_* (glycopeptides)	−17.39	−1.73	1.37	** 1.51 **	0
	−18.25	−11.24	−5.54	** 1.84 **	−2.93
	1.62	1.43	5.78	** 15.89 **	1.11
*lmrS* (oxazolidine, macrolides, phenicols, aminoglycosides)	−1.28	1.77	−1.64	−3.46	2.13
	−74.03	−24.59	−8.11	3.39	−2
	1.42	1.69	16.11	14.62	3.63
*sdrM* (fluoroquinolones)	−1.19	4.69	1.04	−3.07	2.28
	−28.05	−15.67	−20.68	1.97	−1.38
	1	1.09	5.74	32.67	4.23

Green font indicates statistical reproducibility of the results.

## Data Availability

Data are contained within the article.

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
