# Peer review of "A-Type Natriuretic Peptide Alters the Impact of Azithromycin on Planktonic Culture and on (Monospecies and Binary) Biofilms of Skin Bacteria Kytococcus schroeteri and Staphylococcus aureus"

_microorganisms, 2023, doi:10.3390/microorganisms11122965_

Round 1

Reviewer 1 Report

Comments and Suggestions for Authors

Title: A-type natriuretic peptide alters the impact of azithromycin on plank-tonic culture and on (monospecies and binary) biofilms of skin bacteria Kytococcus schroeteri and Staphylococcus aureus.

 The work presents the atrial natriuretic peptide (ANP) alter the effect of azithromycin on Kytococcus schroeteri H01 and Staphylococcus aureus 209P monospecies and binary biofilms. It was demonstrated that the antagonistic effect of the hormone on it occurs mainly on hydrophobic surfaces, whereas an additive effect occurs on hydrophilic surfaces such as glass. Furthermore, ANP, azithromycin, and their combinations caused differential expression of genes of resistance to different antibiotics.

The manuscript fit within the scope of Microorganisms. The document is well written and as required by the journal. I think that the manuscript may be considered for publication on Microorganisms after the minor revision. The comments are as follows:

There were many errors in the manuscript. Please check the whole manuscript.

In the Part of “3.1”, the description of Figure 2E in Sentences 336 and 337 does not match the results shown in the figure.

In the Part of “3.3”, the conclusions obtained in sentences 443 and 444 are inconsistent with the graphical results.

In “Figure 11”, The description of the figure “in the system with forced initial adhesion (B, D, E)” is incorrect, please check.

In line 353 of “Hence, we used 0.65 nM of ANP in the medium.”, this sentence is lack of subject.

Comments on the Quality of English Language

Please improve it. Thanks!

Author Response

Dear Reviewer,

Thank you very much for the valuable remarks. They were truly very useful and allowed us to improve the manuscript. We addressed them, and now we present the revise version of the manuscript.

Reviever. The work presents the atrial natriuretic peptide (ANP) alter the effect of azithromycin on Kytococcus schroeteri H01 and Staphylococcus aureus 209P monospecies and binary biofilms. It was demonstrated that the antagonistic effect of the hormone on it occurs mainly on hydrophobic surfaces, whereas an additive effect occurs on hydrophilic surfaces such as glass. Furthermore, ANP, azithromycin, and their combinations caused differential expression of genes of resistance to different antibiotics.

The manuscript fit within the scope of Microorganisms. The document is well written and as required by the journal. I think that the manuscript may be considered for publication on Microorganisms after the minor revision. The comments are as follows:

There were many errors in the manuscript. Please check the whole manuscript.

In the Part of “3.1”, the description of Figure 2E in Sentences 336 and 337 does not match the results shown in the figure.

Answer. Thank you for the remark. We wanted to say that this inhibitory effect was decreasing in comparison with samples after 24 h and 48 h of incubation. We rewrote the sentence. Now it sounds like “After 72 h of incubation, aninhibitory effect on planktonic cultures of kytococci was also observed, however this effect was not so prominent as after 24 h and 48 h of incubation (fig. 2E). Nevertheless, even a weak….” (lines 393-395).

Reviever. In the Part of “3.3”, the conclusions obtained in sentences 443 and 444 are inconsistent with the graphical results.

Answer. Thank you for the remark. We wanted to say that actually an addition of ANP led to enhance of antibiotic inhibitory effect. We rewrote the sentence. Now it sounds like:

Reviever. In “Figure 11”, The description of the figure “in the system with forced initial adhesion (B, D, E)” is incorrect, please check.

Answer. Thank you. We removed the mistake.

Reviever. In line 353 of “Hence, we used 0.65 nM of ANP in the medium.”, this sentence is lack of subject.

Answer. We rewrote the part of the text. Now it sounds like: “The hormone concentration test was performed previously [16]. Hence, we used 0.65 nM of ANP in the medium because of its effect on the binary biofilms of S. aureus and K. schroeteri as it was demonstrated” (lines 413-416).

Reviewer 2 Report

Comments and Suggestions for Authors

The interactions of microorganisms and their hosts arises enormous amounts of important questions among which - the phenomenon of the interplay of bacteria with the host via its hormones. The authors of the MS examine a complicated system of possible interactions - two microorganisms - S. aureus and K. schroeteri, azitromycin (AZ), and the atrial natriuretic peptide (ANP), in two systems for cultivation of mono- and binary-species biofilms.

The Introduction section marks some keywords related to the topic, unfortunately without detailed and properly subordinated outline of important focus points. This section should be re-ordered, recommendably - re-written, and very much changed to become focused and provide well-ordered arguments that substantiate the objectives of the study.

Methodology is quite sophisticated and in general described in sufficient detail. As a major question - when binary biofilms are analyzed as CFU, the authors should very precisely describe in the methodology section (or elsewhere in the text) how do they discriminate between the colonies of the two bacterial species. Another point is the choice of the wavelength for estimation of bacterial cell density. Why 540 nm instead of the most widely applied wavelengths within the 600-630 range? Minor suggestion: when you use abbreviations, introduce first the whole term, put the abbreviation in brackets and then use it further.

The major problem while reading this MS is the organization of the figures. They are so densely packed that is difficult to follow what has been done and what - not. The same is true for the figure-related texts, that repeat these data without bringing more clarity. This puts questions that might have been addressed by the authors but definitely - not in a reader-friendly way. For instance:

The antibiotic (AZ) is most probably applied in sub-MIC concentrations - what are the MICs for the model strains and how do they refer to the amounts applied in the study? Generally, subMICs of antibiotics by themselves may have diverse effects on biofilm growth - and this is another factor that needs consideration in the results analysis.

Secondly, as noted somewhere in the text, some of the effects may have been provoked by the interaction between the two microorganisms in the binary biofilm. While this seems to be considered by the authors, in the results section this data is not very clearly presented, data on species relation are mixed in the panels together with piles of other bars. Recommendably, more focused attention should be drawn to this interaction - by itself, and not mixed with other data. Of concern, when you analyze binary biofilms by CFU, in some cases you have no positive control value for K. schroeteri - then how do you know you have really a binary biofilm? The explanation that they are difficult to count is not very much scientifically sound. And this finds no help in the CLSM supplemental figures - in spite of the sophisticated approach with the FISH, the images are obscure and not very much readable.

And finally, in order to better support your conclusion on the contribution of the ANP to the observed results, it might be very helpful if you apply a more relevant statistical analysis that could clarify the impact of each of the examined factors - sub-MICs of the antibiotic, inter-species relations between the two strains, mode of cultivation, and finally - the contribution of the host factor (alone or in combination with AZ) to the analyzed parameters.

As a general recommendation: the paper presents interesting results however the organization of the paper - texts and figures, need to be re-worked in way that the statements and conclusions of the authors find clear-cut and unequivocal support.

Author Response

Dear Reviewer,

Thank you very much for the valuable remarks. They were truly very useful and allowed us to improve the manuscript. We addressed them, and now we present the revise version of the manuscript.

Reviever. The interactions of microorganisms and their hosts arises enormous amounts of important questions among which - the phenomenon of the interplay of bacteria with the host via its hormones. The authors of the MS examine a complicated system of possible interactions - two microorganisms - S. aureus and K. schroeteri, azitromycin (AZ), and the atrial natriuretic peptide (ANP), in two systems for cultivation of mono- and binary-species biofilms.

The Introduction section marks some keywords related to the topic, unfortunately without detailed and properly subordinated outline of important focus points. This section should be re-ordered, recommendably - re-written, and very much changed to become focused and provide well-ordered arguments that substantiate the objectives of the study.

Methodology is quite sophisticated and in general described in sufficient detail. As a major question - when binary biofilms are analyzed as CFU, the authors should very precisely describe in the methodology section (or elsewhere in the text) how do they discriminate between the colonies of the two bacterial species.

Answer. Thank you for the remark. We added the information in the beginning of Results section (lines 458-463). K. schroeteri and S. aureus form colonies which are easily distinguished from each other; therefore, the binary community of these bacteria is convenient to analyze for the number of CFU. While K. schroeteri H01 on the solid RCM forms light beige rough, bulging colonies about 2-3 mm in diameter, S. aureus 209P grows in nitidous smooth colonies colored from light yellow to egg yolk color. Also, staphylococcal colonies are normally smaller than kytococcal (1-2 mm in diameter).

Reviever. Another point is the choice of the wavelength for estimation of bacterial cell density. Why 540 nm instead of the most widely applied wavelengths within the 600-630 range?

Answer. Actually, this is a point to discuss a little, because indeed lots of researchers use 600 nm in their studies. However, as we understand, two major arguments are implied here: less damage to microbial cells (in comparison with UV) and less absorbance by growth media (media for saprotrophic bacteria are frequently colored in from yellowish to orangish/brownish shades). Some studies (for instance https://www.liebertpub.com/doi/epdf/10.1089/pho.2012.3343) also demonstrated even a slight bactericidal effect of 525 nm diode on some bacteria (however it depended on light intensity etc.). And 625 nm in that study was established as not bactericidal. However, to be really dangerous for bacteria. The >500 nm light should be very intense and it is impossible to kill or damage bacteria in the main suspension while only a part of a suspension is taken to be measured for its OD and then discarded. In case of kinetic studies, the intensity of light seems to be also not sufficient enough to kill bacteria (at least in comparison with the light used in the study mentioned above).

Next, as it was studied before (https://www.science.org/doi/pdf/10.1126/science.121.3151.709?casa_token=E4Nd5KmzPl4AAAAA:lpGTt3_YG90vYfzmRVYbrun3MGt3mYfiSNnlG2nrmCPcS-X3AsZ7Y7jPPwXODOlAbhDuOhJqmkmSsA; https://link.springer.com/article/10.1007/BF01499867; https://escholarship.org/content/qt9fp494sg/qt9fp494sg.pdf) different bacterial species have different absorption spectra which depend on the pigments, cell shape etc. And sometimes 600 nm absorption is rather low, lower than for instance at 540-550 nm (or at other wavelengths). Also, to go deeper in the “OD600 measurements”, some studies were devoted to provide an additional corrections and normalizations (as it was applied here https://www.ncbi.nlm.nih.gov/pmc/articles/PMC5102515/#:~:text=Based%20on%20optical%20spectroscopy%2C%20an,resulting%20in%20the%20nomenclature%20OD600). However, in this article, absorption at 544 nm was not critically different from the absorption at 600 nm, and the curves were of identical shape.

Hence, we do suppose that both wavelengths 540 nm and 600 nm (or any other wavelength between 540 and 600 nm) are appropriate for suspension OD measurements. In our lab the use of 540 is a part of “tradition”. When we started our works, the filter 540 nm was the most appropriate among other to measure the cell suspension ID. When we grew up and get bigger machinery and research facilities, we continued measurements at 540 nm to get to maintain continuity and logical connection between previous and “to date” works. When I worked in France in Rouen University, the lab there had its own tradition – they measure the OD at 580 nm (https://www.mdpi.com/2076-2607/10/9/1788, https://www.sciencedirect.com/science/article/pii/S259020752300028X, https://www.frontiersin.org/articles/10.3389/fmed.2019.00155/full, https://www.frontiersin.org/articles/10.3389/fmicb.2018.02912/full). As for 540 nm – here there three articles that were found in Google less than in 2 minutes where OD540 was measured: https://www.ncbi.nlm.nih.gov/pmc/articles/PMC261101/pdf/iai00106-0310.pdf, https://pubmed.ncbi.nlm.nih.gov/14742208/, https://www.sciencedirect.com/science/article/pii/S1049964409000474).

Hence, we do suggest that no significant differences in results can be occurred after use of 540 nm wavelength instead of mainstream 600 nm. The main requirement is to use an adequate control and process al the samples in the same manner.

Reviever. Minor suggestion: when you use abbreviations, introduce first the whole term, put the abbreviation in brackets and then use it further.

Answer. We checked and made the corrections.

Reviever. The major problem while reading this MS is the organization of the figures. They are so densely packed that is difficult to follow what has been done and what - not. The same is true for the figure-related texts, that repeat these data without bringing more clarity.

Answer. Thank you for the remark. Indeed, we had to put a lot of data in the figures. However, we tried to make them more visible and to enlarge the describing texts as you recommended where it was possible.

Reviever. This puts questions that might have been addressed by the authors but definitely - not in a reader-friendly way. For instance:

The antibiotic (AZ) is most probably applied in sub-MIC concentrations - what are the MICs for the model strains and how do they refer to the amounts applied in the study?

Answer: We found some information about the MICs.

  1. aureus 209P - Erythromycin 30-32 µg/mL (https://bacdive.dsmz.de/strain/14445). In our case, we suppose the MIC of azithromycin is between 10-15 µg/ml.
  2. schroeteri – no data were found in public strain collections. In one article (https://www.ncbi.nlm.nih.gov/pmc/articles/PMC3264184/) there was the MIC 12 µg/ml of erythromycin for K. schroeteri strain isolated from paien with pneumonia. Our strain seem to have the MIC of azithromycin between 40-50 µg/ml of azithromycin (we tested it specially some days ago).

Reviever. Generally, subMICs of antibiotics by themselves may have diverse effects on biofilm growth - and this is another factor that needs consideration in the results analysis.

Answer. Yes, thank you for the remark. We added the information and explanation in the text (lines 333-343):

First, we selected work concentrations of azithromycin for further experiments. Because of necessity to investigate simultaneous effects of azithromycin and ANP and interactions between S. aureus and K. schroeteri, we decided to take concentrations with statistically significant but quantitatively lower effects. The values should provide an ability to find any changes caused by an addition of the hormone or presence of another microorganism in the binary biofilm. Hence, the search should be conducted mostly in range of subinhibitory concentrations. Also, in lower concentrations antibiotics may play a signal role and regulate the bacterial behavior (Yim et al., 2007; Romero et al., 2011). Next, subinhibitory concentrations are of special interest because of their frequent appearance during incorrect chemotherapy of infections. Hence, in lower concentrations, antibiotics should be studied as well as in traditional higher concentrations.

Reviever. Secondly, as noted somewhere in the text, some of the effects may have been provoked by the interaction between the two microorganisms in the binary biofilm. While this seems to be considered by the authors, in the results section this data is not very clearly presented, data on species relation are mixed in the panels together with piles of other bars. Recommendably, more focused attention should be drawn to this interaction - by itself, and not mixed with other data. Of concern, when you analyze binary biofilms by CFU, in some cases you have no positive control value for K. schroeteri - then how do you know you have really a binary biofilm? The explanation that they are difficult to count is not very much scientifically sound. And this finds no help in the CLSM supplemental figures - in spite of the sophisticated approach with the FISH, the images are obscure and not very much readable.

Answer. Thank you for the remark. We tried to rewrite the text and add some phrases about the interactions between bacteria as you recommended.

Reviever. And finally, in order to better support your conclusion on the contribution of the ANP to the observed results, it might be very helpful if you apply a more relevant statistical analysis that could clarify the impact of each of the examined factors - sub-MICs of the antibiotic, inter-species relations between the two strains, mode of cultivation, and finally - the contribution of the host factor (alone or in combination with AZ) to the analyzed parameters.

Answer. We used the Mann-Whitney U-test as it was noted in the text. However it is tricky sometimes to clarify the impact of each factor, hence we decided to take one control – samples without additions. We first tried to redraw images and use additional controls to “isolate” each compound. For instance, we used 0.001 µg/ml azithromycin as a control for 0.001 µg/ml + ANP samples etc. However, we declined this approach because of (i) the amount of figures become truly enormous even in comparison with existing one. And (ii) there was no changes in statistical significance when we calculated for instance relative values using samples with antibiotic as controls for combinations in comparison with using no-additions samples as universal controls. Yes, maybe sometimes (only sometimes) some diagrams became a little more demonstrative, however it was not universal. So, we ask for the for kindly giving us the opportunity to leave the images as they are to avoid too many images.

Concerning the parameters for statistical analysis, unfortunately we cannot compare everything. For instance, we use two model systems and we actually cannot compare results because they are too far ones from others. Actually. The longer we work with bacteria-hormone models, the more we understand that (i) we do know nothing, and (ii) actually AI support is needed. Because we can use some statistics of course, use q-values instead of p-values and FDR corrections, use ANOVA (I have tested it for some random samples groups and it demonstrated mostly p<0.0001), however truly

Also, one of the main tricky things here is that hormones mostly affect bacteria “invisibly”. We can find changes 10-20% in biofilm growth for instance (measured with crystal violet) or no changes at all. However when we go “deeper” and study for example proteomic changes we can find a lot of alterations. We met such phenomena in most cases we work with hormones. For instance, in our last article dedicated to proteomics of M. luteus or in the article about the ANP effect on S. aureus and K. schroeteri. So, hormones act in multitarget manner. And this must also be analyzed, but what an approach is applicable here – actually I still do not know the certain answer.

Reviever. As a general recommendation: the paper presents interesting results however the organization of the paper - texts and figures, need to be re-worked in way that the statements and conclusions of the authors find clear-cut and unequivocal support.

Answer. We tried to make images more “bright” and added more visual support. Also we added more statistical analysis. We are also added more explanation in the text. Thank you for the remark.

Round 2

Reviewer 2 Report

Comments and Suggestions for Authors

The revised text has been significantly improved, clear and readable, and is in accordance with the requirements for publication in Microorganisms.